# Kinomics platform using GBM tissue identifies BTK as being associated with higher patient survival

Sofian Al Shboul[1,8] , Olimpia E Curran[2,3], Javier A Alfaro[1,4], Fiona Lickiss[4], Erisa Nita[1], Jacek Kowalski[4], Faris Naji[5], Rudolf Nenutil[6], Kathryn L Ball[1] , Radovan Krejcir[6], Borivoj Vojtesek[6] , Ted R Hupp[1,4], Paul M Brennan[1,7]

**Better understanding of GBM signalling networks in-vivo would help develop more physiologically relevant ex vivo models to support therapeutic discovery. A "functional proteomics" screen was undertaken to measure the specific activity of a set of protein kinases in a two-step cell-free biochemical assay to define dominant kinase activities to identify potentially novel drug targets that may have been overlooked in studies interrogating GBM-derived cell lines. A dominant kinase activity derived from the tumour tissue, but not patient-derived GBM stem-like cell lines, was Bruton tyrosine kinase (BTK). We demonstrate that BTK is expressed in more than one cell type within GBM tissue; SOX2-positive cells, CD163-positive cells, CD68-positive cells, and an unidentified cell population which is SOX2-negative CD163-negative and/or CD68-negative. The data provide a strategy to better mimic GBM tissue ex vivo by reconstituting more physiologically heterogeneous cell co-culture models including BTK-positive/negative cancer and immune cells. These data also have implications for the design and/or interpretation of emerging clinical trials using BTK inhibitors because BTK expression within GBM tissue was linked to longer patient survival.**

## Introduction

Glioblastoma (GBM), a World Health Organization (WHO) grade IV astrocytoma, is the most common aggressive tumour originating in the brain (1). The incidence is 4.64/100,000 per year (2). Median survival is less than 15 mo, even with surgery, radiotherapy and chemotherapy using the alkylating agent temozolomide (TMZ) (3, 4). Immune evasion, drug resistance, invasion of surrounding brain tissue, intratumoural heterogeneity, necrosis, and aggressive vascularization have all been implicated in the poor response of GBM to

treatment (5, 6, 7, 8, 9, 10). Efforts to identify novel effective therapeutic targets have largely failed, with few drugs successfully translating from drug trials into standard clinical use, and no significant impact on patient survival (11, 12, 13). For example, bevacizumab, a VEGF monoclonal antibody (14, 15), in combination with TMZ and/or lomustine improves progression-free, but not overall survival (16, 17). Novel therapeutic targets are urgently needed.

Over the past 30 yr, protein kinases have been the most widely therapeutic target in all cancers (18). In GBM, receptor tyrosine kinase (RTK) gene alterations occur at high frequency, for example, more than half of GBMs showed evidence of epidermal growth factor receptor (*EGFR*) aberrations (19), likewise, platelet-derived growth factor receptor (*PDGFR*) and vascular endothelial growth factor receptor (*VEGFR*) were found in 40% and 39% of examined GBMs, respectively, making them attractive potential targets for therapeutic inhibition (20, 21). To date though, drugs against kinase targets have not been clinically effective in GBM. For example, drugs targeting EGFR/VEGFR and EGFR/HER2, such as vandetanib and lapatinib, respectively, showed little effect on the overall survival of patients (22, 23, 24). Sunitinib, an inhibitor of VEGFRs, PDGFRs, c-KIT, FLT3, and the RET kinases, failed to make an impact as monotherapy for non-resectable or for recurrent GBMs (25, 26). Cediranib, a kinase inhibitor against VEGF, and enzastaurin, an inhibitor of PKC and PI3K/AKT pathways, also failed in recurrent GBM (27, 28). Therapy failure may be a consequence of RTK intratumoural heterogeneity (29, 30, 31) or may reflect that these RTKs could have various functions/roles in tumour tissue which could be lost/altered upon deriving cell lines because of culture conditions.

The failure of preclinical drug discovery to identify compounds in the laboratory that translate effectively as drugs into the clinic in part reflects that the in-vitro models used for compound screening do not adequately recapitulate the complexity of the cellular, immune, and microenvironment interactions necessary for tumour growth and survival, or the physiological conditions under which these interactions take place. It is even unclear to what extent human GBM-derived cell lines mimic and resemble the tumour

[1]Institute of Genetics and Cancer, University of Edinburgh, Edinburgh, UK  [2]Department of Neuropathology, Western General Hospital, Edinburgh, UK  [3]Cardiff University Hospital, Cellular Pathology, Cardiff, UK  [4]International Centre for Cancer Vaccine Science, University of Gdansk, Gdansk, Poland  [5]Pamgene International BV, 's-Hertogenbosch, Netherlands  [6]Research Centre for Applied Molecular Oncology, Masaryk Memorial Cancer Institute, Brno, Czech Republic  [7]Translational Neurosurgery, Centre for Clinical Brain Sciences, University of Edinburgh, Edinburgh, UK  [8]Department of Basic Medical Sciences, Faculty of Medicine, The Hashemite University, Zarqa, Jordan

Correspondence: s.alshboul@ed.ac.uk; paul.brennan@ed.ac.uk

cells in the primary tissue at the proteome level. For example, Vogel and colleagues compared four serum-grown GBM cell lines against eight non patient-matched GBM tissue samples and reported more than 300 proteins were gained or lost in the cell lines (32). GBM-derived cells cultured in serum conditions differ dramatically from those cultured in the serum-free media conditions that favour survival of the cancer stem-like cells implicated in GBM growth and therapy resistance (33), which may explain the differences Vogel observed.

We assessed dominant kinase activity in whole GBM tissue lysates to identify potentially novel drug targets that may have been overlooked in studies interrogating GBM-derived cell lines. A "functional proteomics" screen was undertaken to measure the specific activity of a subset of protein kinases in a two-step cell-free biochemical assay to define dominant kinase activities in tissue and whether cell line models mimic these effects. In contrast to a proteomics screen that measures steady-state protein level, the kinomics screen measures the specific activity of an enzyme. A dominant kinase activity in the tumour tissue was Bruton tyrosine kinase (BTK). BTK inhibitors are currently being used in GBM clinical trials (NCT03535350). Because BTK is known to be expressed in both cancer cells and immune cells (34, 35, 36, 37), we focused on defining the cell types within GBM tissue where BTK is expressed. We demonstrate that BTK is expressed in at least four cell types; SOX2-positive cells, CD163-positive cells, CD68-positive cells, and a fourth cell population which is SOX2 negative, CD163 negative, and/or CD68 negative. The data provide support for a strategy to better

mimic GBM tissue ex vivo by reconstituting more physiologically heterogeneous cell co-culture models that include BTK-positive/negative cancer and immune cell populations. These data also have implications for the design and/or interpretation of emerging clinical trials using BTK inhibitors because BTK-positive expression within GBM tissue was linked to longer patient survival.

# Results

## Cell-free kinase assays reveal kinome differences between GBM cancer tissue and patient-matched cell lines

The functional status of protein kinases in GBM tissue and patient-matched tumour-derived cell lines was measured using a Pamstation12 kinomics platform (Fig 1). The platform contains an array of 194 phosphoacceptor substrates whose phosphorylation rate can be measured using fluorescently labelled phosphoantibodies. The kinome profile was measured in fresh frozen GBM tumour tissue from two different patients (G-221 and G-222) and the matched primary cell cultures derived from adjacent pieces of tissue. Cell-free kinase activity was processed on the spot arrays to measure phosphorylation rates, at different exposure times of the same chip (Fig 2A). Both tissue and matched glioblastoma stem-like cell liners derived from the same resected tumour tissue were used to additionally determine whether our ex vivo models mimic the

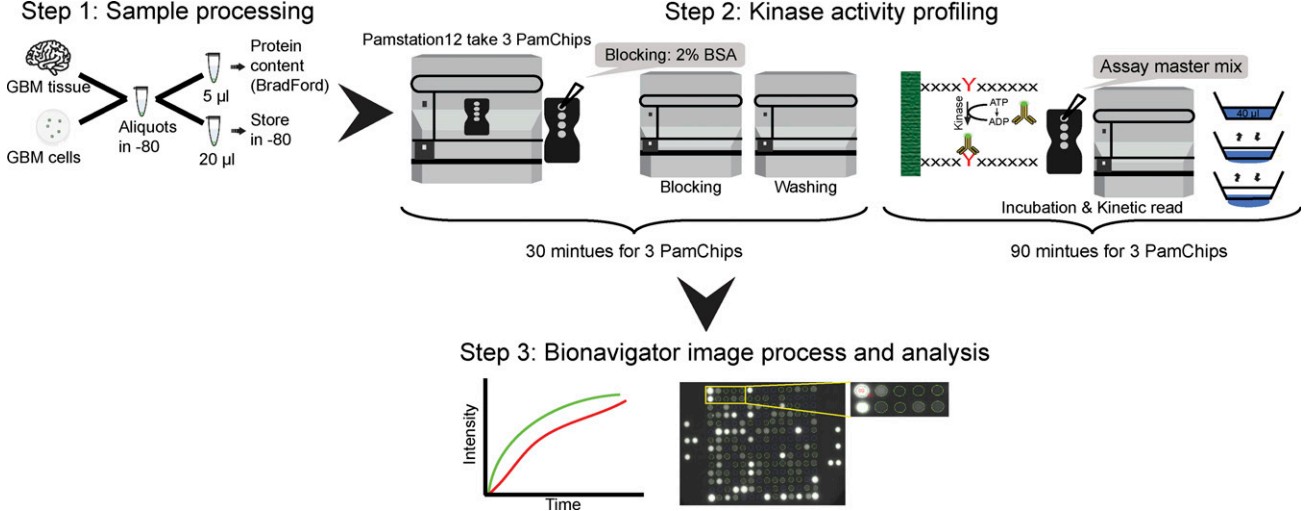

**Figure 1.   Schematic diagram of kinomics analysis using the Pamstation12 platform.**
(Step 1: Sample processing). Surgically resected GBM tissue and patient-matched GBM cell lines derived from adjacent tumour tissue were aliquoted and stored at −80. After cancer tissue or cell line lysis using kinase lysis buffer, which preserves bioactive enzymes, the soluble protein concentration was quantified using the Bradford method and fixed amounts of total protein from tissue or cell lines was processed on the PamChips. (Step 2: Kinase activity profiling). After 30 min of blocking the PamChip with 2% BSA, the kinase reactions were assembled in a buffer (40 µl) containing 5 µg of total soluble protein, fluorescently labelled phospho-specific antibody, and ATP. The reactions were loaded in each of the four wells of a PTK PamChip containing 194 synthetic peptides and incubated for 30 min at room temperature. The vertical movement of the mixture solution through a porous ceramic membrane drives the fluorescently labelled antibody to bind to phosphorylated peptide substrates. (Step 3: Bionavigator image process and analysis). A charge-coupled device camera placed within the Pamstation12 platform captures images as a function of time to produce kinetic data (Vini) that measures phosphorylation rates of each peptide as well as end point data (EP) that measures the total peptide phosphorylation over the time course. Using Bionavigator6, a grid is fitted on each image to calculate the differences between the signal (SG) and background (BG) intensities to produce kinetic values for each substrate. The raw values are converted to the highest change of fluorescence for each cycle to define the mostly linear rate of phosphorylation and unsupervised hierarchical clustering and statistical tests are performed. The spot array images are from a representative sample were acquired using the charge-coupled device camera and were prepared and processed using Bionavigator.

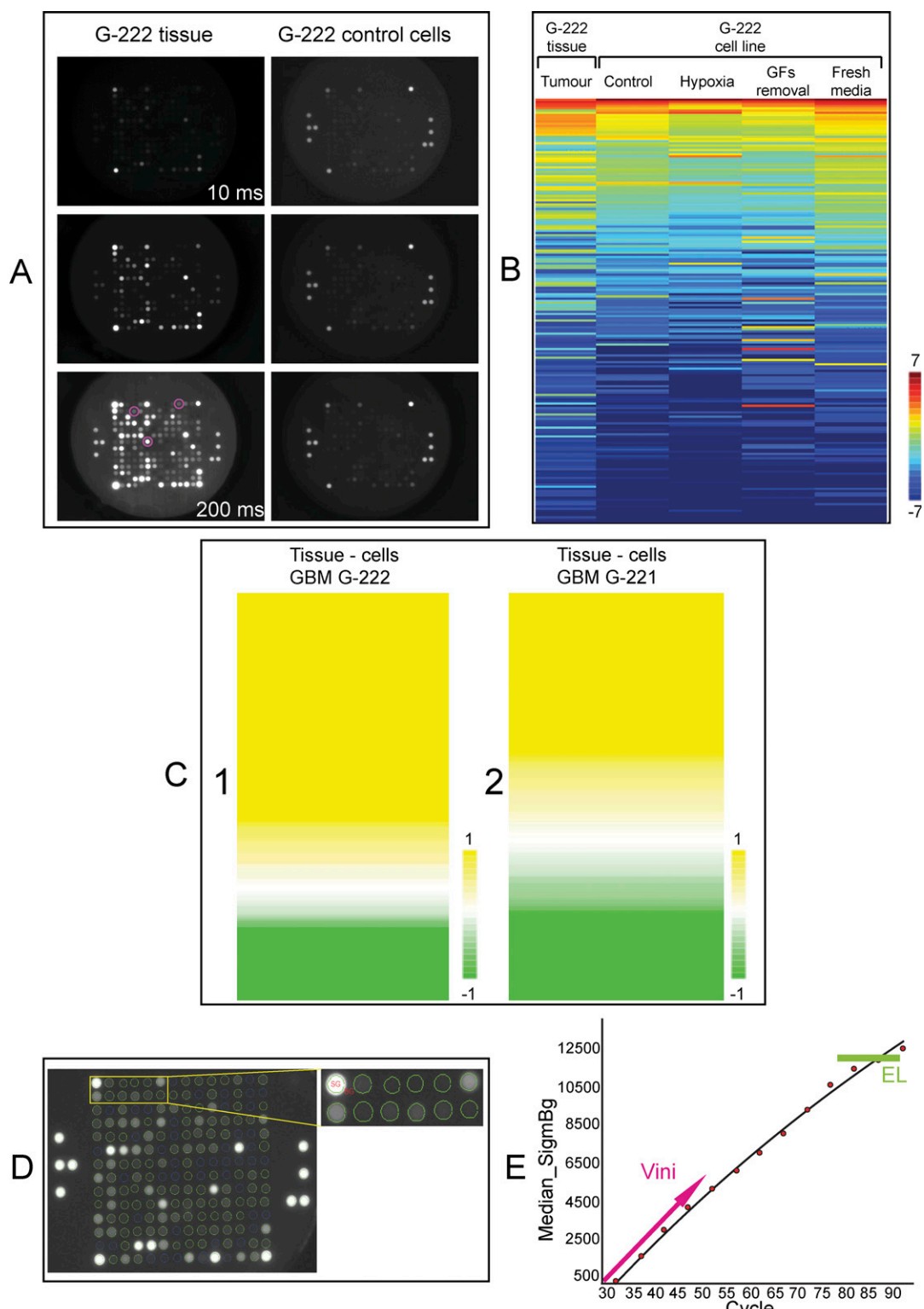

**Figure 2. Comparison of the kinome activity in lysates derived from G-222 GBM tissue and matched GBM cell line.**
**(A)** Spot array images at different exposure times of the same sample (10, 50, and 200 ms) for the same protein input using lysates from either G-222 tumour tissue or the G-222 stem-like cell line (control) derived from the same patient. The pink circles highlight representative peptides that exhibits elevated phosphorylation using lysates from G-222 tissue relative to the G-222 cell line. **(B)** A rainbow heat map defining the phosphorylation status of the 194 peptide substrates using all G-222 stem-like cell derived lysates (under the indicated conditions of normal media, hypoxia mimetic, growth factor removal, and addition of fresh media) versus the G-222 GBM tissue lysate. The peptides were ranked according to their phosphorylation status; top-ranked are in the red spectrum (+7) and the peptides with lower rates of phosphorylation are highlighted in the blue spectrum (−7). **(C)** Colored heat map that measures changes in the peptide phosphorylation status: (1) the ratio between three biological

physiological state of the GBM tissue with respect to dominant protein kinase activity.

In patient G-222, the total kinase activity (specific activity) was substantially higher in lysates derived from GBM tissue relative to the patient-matched cell line (Fig 2A, tissue versus stem-like cell line panel). This is visually evident when the data from the same chip is collected at 10, 50, or 200 ms exposure times where the spot intensity increases substantially over the time course using GBM tissue as a source of kinase (Fig 2A; G-222 Tissue panel). The circled spots (200 ms panel) highlight some representative phospho-peptides that are elevated in tissue relative to the stem-like cell line. By contrast, when the data using the same chip are measured using G-222 stem-like cell line lysates as a source of kinase, the phosphopeptide signal is visually diminished (Fig 2A; G-222 control cell lysate panel). These data indicate that cell lines and matched tissue do not share dominant kinomes. This might reflect inherent differences in the tumour cell proteome between the in-vitro and in-vivo tissue states, perhaps driven by differences in oxygen tension and nutrient availability in-vivo compared with in-vitro. Tumour tissue is also a composite of many different cell types, so these differences may alternatively be explained by the non-tumour cell kinomes. Both scenarios may be operating.

To investigate these possibilities we first altered the in-vitro treatment conditions of the glioblastoma derived G-222 stem-like cell line to explore a spectrum of possible GBM microenvironments that might induce kinases in the cell lines to better reflect the GBM tissue kinome. G-222 tissue was compared to G-222 cell line under different conditions (Fig 2B); (i) standard media with hypoxic conditions chemically mimicked, (ii) standard media with EGF and FGF growth factors removed, and (iii) media and oxygen conditions with regular fresh media exchange to remove autocrine factors which essentially renews the cells with high glucose containing media. The rate of phosphorylation (Vini) for each of the culture conditions is presented in a rainbow heat map of the 194 peptide substrates in lysates derived from G-222 stem-like cells and G-222 derived GBM tissue (Fig 2B). The heat map data show that each change (hypoxia, growth factor removal, fresh media addition versus control panel) induces a different bar code pattern which is consistent with a change in the kinome by each treatment; however, none of these conditions creates a bar code that looks similar to the tumour tissue kinome bar-coded pattern (Fig 2B).

Next, we performed a comparison between three replicates of G-222 control GBM stem-like cells and three replicates of G-222 tissue and we found that 142 peptides (142/194) were higher in G-222 tissue (Fig 2C1, yellow color), whereas only 52 peptides exhibited higher activity in G-222 control cells (Fig 2C1, green color).

We then analyzed a second pair of tumour-derived stem-like cell lines and patient-matched tumour tissue from a different patient, G-221. Two biological replicates of lysates from G-221 GBM tissue and three replicates of matched G-221 tissue-derived stem-like cell lines were processed using the same kinomics platform. Similar to the material from G-222, samples from G-221 exhibited some discordance in the kinase profiles between the tissue-derived and cell line–derived lysates (Fig 2C). A total of 119 (119/194) peptides exhibited higher phosphorylation activity within G-221 tissue compared with G-221 control cells (Fig 2C2, yellow color), whereas 75 peptides were higher in G-221 control cells (Fig 2C2, green color). The rainbow heat maps generated using rate values determined over early time points when the kinase reaction was mostly linear (Vini), rather than end point values which would have instead given only a final total amount of substrate phosphorylated (Fig 2D and E). Together these data suggest that the tissue kinome and GBM stem-like cell kinomes are distinct.

A second analysis of the kinome activity was undertaken to assess further the apparent discordance in kinome activity between tissues and cell lines and to determine which kinases were significantly different. The kinome activity quantified using the Vini data (early rate of the reaction) was analyzed by unsupervised hierarchical clustering with a *t* test, using the three biological replicates of the G-222 tissue and of the G-222 cell line grown under standard conditions (Fig 3A). A total of 119 peptides (119/194) with a *P*-value < 0.05 were identified (Tables S1 and S2). These data were viewed in a volcano plot to assess the peptides that were differentially phosphorylated using lysates derived from G-222 GBM tissue or from the G-222 cell line (Fig 3B). These data confirmed the earlier observations that CNTB1, PDGFRB, and CBL (Fig 3B, blue circles) exhibited higher phosphorylation rates in lysates from GBM tissue, whereas EGFR and INSR (Fig 3B, green circles) exhibited higher levels of phosphorylation in lysates from the matched stem-like cell line. Brown circled peptides (Fig 3B) are proposed to be interacting with BTK according to the String map (see below; Fig 4B).

### BTK emerges as a dominant bioactive kinase in GBM tissue

After establishing a significant difference in the kinome activity between lysates from G-222 GBM tissue and from the patient-matched cell culture, we stratified the top 10 active peptides in the G-222 tumour tissue to determine whether there were any kinases represented that were previously linked to biologically relevant pathways. The top kinase that emerged was BTK. This kinase has a role in the upstream/downstream signalling pathway of the six highest expressed phosphoproteins identified in lysates

---

replicates of the G-222 GBM tissue lysate versus three biological replicates of G-222 control cells for all the 194 peptides. 142 peptides exhibited higher phosphorylation using G-222 tissue comparing to stem-like cell line (yellow color; +1), whereas 52 peptides exhibited higher phosphorylation in cell line versus tissue (green color; −1), and (2) the ratio between two biological replicates of lysates derived from the G-221 GBM tissue versus three biological replicates of G-221 derived stem-like cells for all the 194 peptides. 119 peptides exhibited higher phosphorylation using G-222 tissue comparing to stem-like cell line (yellow color; +1), whereas 75 peptides exhibited higher phosphorylation in the stem-like cell line versus tissue (green color; −1). **(D, E)** Phosphorylation activity plots highlighting the linear relationship between end-level and Vini values along with spot array images with a grid used to calculate initial kinetic values. **(D)** Each image taken throughout the measuring process is fitted with a grid to calculate the delta between median fluorescent signal (SG) and median background at 100 times the change of signal in 1 ms estimated using several exposure times. These images were acquired by the charge-coupled device camera using representative control lysates and were prepared and processed using Bionavigator. **(E)** The linear relationship between end-level and Vini values suggesting using either value will produce the same results (credit: Pamgene international). However, in our analysis, we use the early rate reaction at the early time point which is standard practice in measuring the specific activity of an enzyme.

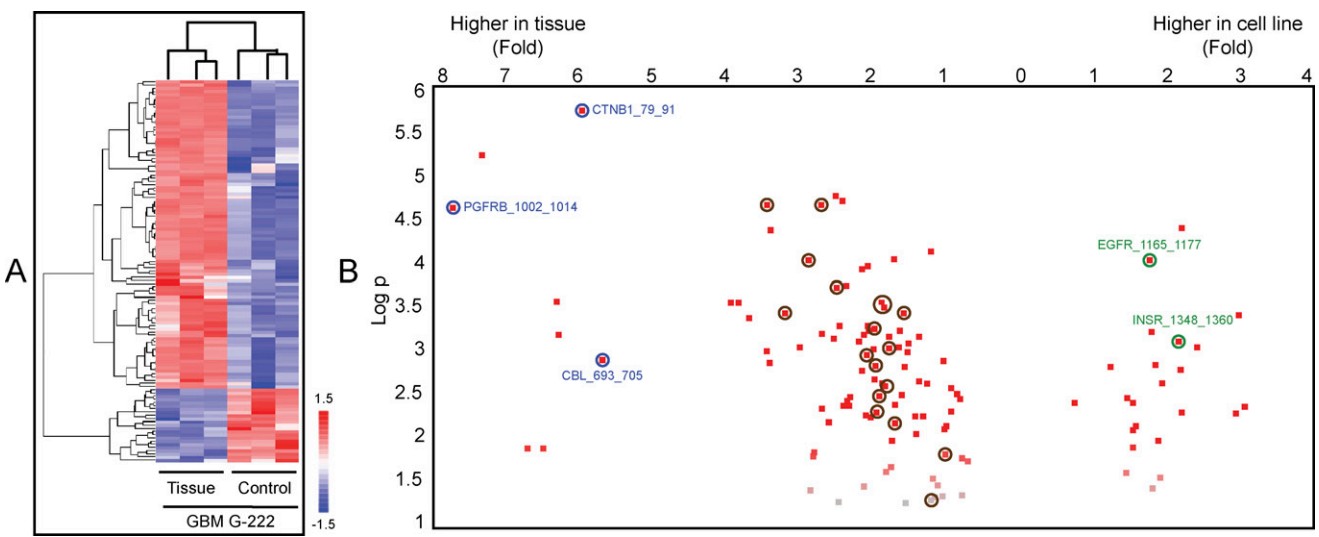

**Figure 3. Orthogonal quantitation of kinase activity in lysates derived from tissue and cell lines.**
**(A)** A two-color heat map using unsupervised hierarchical clustering and a *t* test with three biological replicates of lysates derived from either G-222 tumour tissue or the G-222 stem-like cell line grown under standard conditions identified 119 peptides (119/194) with a *P*-value < 0.05 (list of peptides in Tables S1 and S2). Red indicates higher peptide phosphorylation (+1.5), whereas blue indicates lower phosphorylation (−1.5). **(B)** Volcano plot for the *t* test between G-222 tumour tissue and the G-222 cell line, peptides are presented as dots. The x-axis values show the fold difference in peptide phosphorylation between G-222 tissue and G-222 cell line. The y-axis shows the log of *P*-value (>1.3 was considered statistically significant). Blue circled peptides were phosphorylated more than fivefold higher in lysates derived from cancer tissue compared to stem-like cell line, whereas green circled peptides were phosphorylated more than twofold higher using lysates derived from the G-222 stem-like cell line versus G-222 tissue. Brown circled peptides are proposed to be interacting with Bruton tyrosine kinase according to the string map shown in Fig 4B.

from the G-222 tissue (CD79A, PTN6, PECA1, PLCG1, ZAP70, and PLCG2) (38, 39, 40, 41, 42, 43). Further evaluation of additional proteins known to be either phosphorylated by BTK or to interact with BTK, revealed another 12 peptides within the top 100 active peptides with different tyrosine sites that are BTK-related targets (PTN6, PECA1, ZAP70, PTN11, P85A, EPOR, KIT, LAT, LCK, and JAK2) (Fig 4A). Furthermore, we performed the same analysis using the same 18 peptides on the G-221 GBM tissue and control G-221 stem-like cell line and we found that all the 18 peptides were higher in G-221 tissue relative to the G-221 cell line (Fig S2). All of the parental proteins of these 18 phosphopeptides interact with BTK and are shown in the String map (Fig 4B). Furthermore, of 119 differentially modified peptides passing the *t* test that were identified in lysates derived from G-222 tissue, 18 are from proteins known to interact with BTK (Table S3). In addition, these same 18 peptides also exhibit higher phosphorylation states using the kinomics platform, when comparing G-221 tissue versus G-221 derived cell line (Fig S3).

To corroborate the observations that BTK-associated activity was higher in lysates derived from GBM tissue, we performed a kinome analysis of lysates from a further 31 distinct GBM tissue samples taken from 17 patients (Figs 4C and S2). The tissues were lysed, and the cell-free kinase assay was processed on the spot arrays to measure phosphorylation rates as above. The Vini rate data were processed in a heat map using unsupervised clustering (Fig 4C). The top 10 phosphopeptides identified included the same BTK pathway targets (Fig 4C); CD79A, PLCG1, PTN6, PECA1 ZAP70, and PLCG2 (38, 39, 40, 41, 42, 43), reinforcing the relevance of elevated BTK-associated kinase activity in lysates derived from GBM tissue.

We next used immunoblotting to specifically evaluate BTK protein expression in a panel of 29 GBM tumour tissue samples

taken from 18 patients (patients 604, 376, 563, 1,285, 607 and 581 were sampled from multiple regions of each tumour) (Fig 5A), 10 GBM stem-like cell lines grown in serum-free media (GCGR-E20, GCGR-E21, GCGR-E22, GCGR-E25, GCGR-E27, GCGR-E28 GCGR-E34, GCGR-E56, GCGR-E64, and GCGR-E65) and one GBM cell line grown in serum-containing media (U251) (Fig 5B). We included two human neural cell lines as "normal brain" controls (GCGR-NS12ST_A and GCGR-NS17ST_A), grown in serum-free media (Fig 5B). BTK protein was expressed in the lysates of 28 of 29 GBM tissue samples (Fig 5A), consistent with the high BTK-associated kinase activity in the cell-free assay. In all of the GBM stem-like cell lines, BTK protein was below the level of detection (Fig 5B), consistent with the relatively low BTK-associated activity from G-222 cell line (Figs 3 and 4). The immunoblotting data are thus consistent with the kinomics platform data; higher BTK-associated kinase activity occurs in GBM tissue lysates under conditions in which BTK protein can be detected, whereas lower BTK-associated activity occurs in GBM-derived cell lysates under conditions in which BTK protein is not detected.

In GBM cell lines and mice xenografts, BTK silencing or inhibition results in reduced neurosphere forming ability, migration, and glioma cell proliferation, and induces apoptosis (37, 44, 45). This suggests that BTK is pro-oncogenic and consequently BTK is considered a drug target in GBM (35, 37, 44, 45). The evidence for oncogenic data was derived from GBM cell lines grown in serum rich media (37, 44, 45), conditions known to promote differentiation of the stem-like cells characteristics of GBM cells (33). Glioma stem-like cells have been implicated in GBM development and therapy resistance. Our analysis of the G-222 GBM stem-like cell line (grown in serum-free media to maintain stem-like features) demonstrated

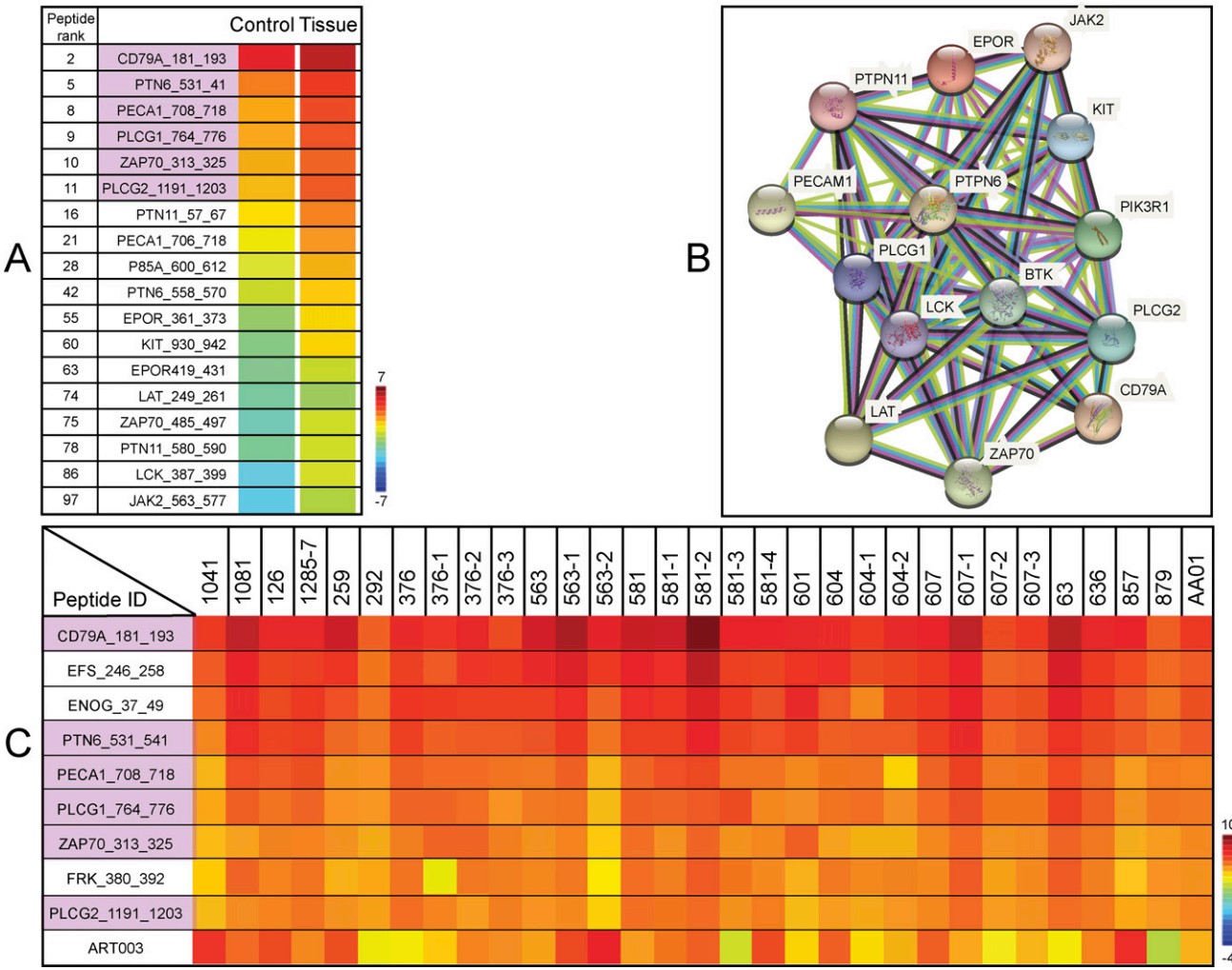

**Figure 4. Kinomic analysis highlighting Bruton tyrosine kinase (BTK) kinetic activity based on peptide phosphorylation using GBM tissue and stem-like cell lysate samples.**

**(A)** Rainbow heat map for kinase activity in lysates derived from G-222 tumour tissue and the control G-222 stem-like cell line. The 18 peptides illustrated are predicted phosphorylation targets of BTK (www.phosphonet.ca) or interact with BTK. Six of these 18 were among the top 10 active peptides (pink highlighted), whereas the other 12 were found within the peptides ranked from 12 to 100. Peptides were ranked according to their phosphorylation activity; higher rank indicates higher phosphorylation rate (peptide ranks). **(B)** Protein interaction map built using String (https://string-db.org/) showing the parental proteins of the peptides that have a potential interaction with BTK. **(C)** Rainbow heat map illustrating the top 10 active peptides in lysates from 31 GBM tissues (Full heat map of all the peptides in Fig S1). Tumour sample IDs with the same three or four digit prefix (i.e., 563, 563-1, and 563-2) come from the same patient, but from different regions of the same tumour tissue. The heat map was generated using Vini values (see the Materials and Methods section) and peptides were arranged based on the highest rate of phosphorylation activity (peptide rank). Red indicates higher rates of phosphorylation activity (+10) and blue lower phosphorylation activity (−4). The pink color in the "peptide ID column" highlights peptides; CD79A, PLCG1, PTN6, PECA1 ZAP70, and PLCG2, which are suggestive of BTK activity (38, 39, 40, 41, 42, 43), thus illustrating the activity associated with BTK within different GBM tissues. The synthetic pre-phosphorylated peptide named ART003 (EAI(phospho-Y)AAPFAKKKXC) (bottom of the row in Fig 4C) is a non-physiological control phosphopeptide incorporated by the Pamstation12 user platform that functions as a positive control as it exhibits a high basal signal because of its prior phosphorylation. As this phosphopeptide is "100%" phosphorylated without any lysate addition, any reductions in its phosphorylation state (such as indicated by the green color in tissue 879) is likely due to phosphatase(s) from the tissue that de-phosphorylates the peptide. Data from this phosphopeptide were not used in any pathway assessments as it is only used as a positive control to yield a high phosphosignal in the Pamstation12.

that by comparison to the patient-matched tissue, BTK-associated activity was actually low in the cell lines (Figs 3 and 4), making it less likely to be pro-oncogenic. To interrogate this further, we assessed BTK protein expression in two different GBM stem-like cell lines derived from matched tissue, using immunoblotting (GCGR-E27 and GCGR-E28) (Fig 6). In both cell lines, BTK protein was below the level of detection (Figs 6A and D and S4). By contrast, in the matched GBM tumour tissue, including patient-matched tissue for cell lines GCGR-E27

(Fig 6A versus Fig 6B and C) and GCGR-E28 (Fig 6D versus Fig 6E and F), BTK protein expression was clearly present.

## BTK expression in a glioma tissue microarray

The apparently discordant expression we observed between BTK protein expression in GBM-derived stem-like cell lines and tumour tissue is important to resolve because BTK is presumed to be

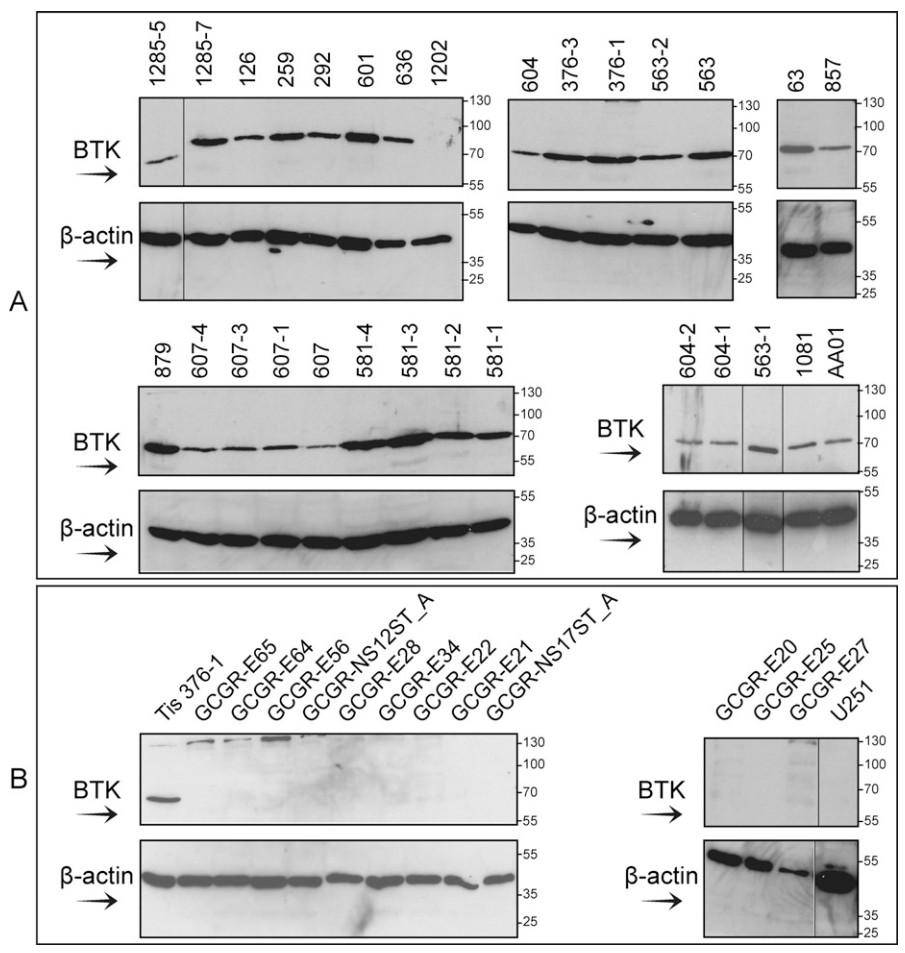

**Figure 5. Immunoblotting analysis of Bruton tyrosine kinase (BTK) protein expression in 29 GBM tumour tissues, 11 GBM cell lines, and two human brain fetal cell lines.**

**(A)** GBM tissue samples were lysed and processed for immunoblotting with antibodies to BTK or actin, as indicated. The numbers indicate the tumour code used for archiving. BTK protein expression was detected in all GBM samples except one (28/29), sample #1202 BTK was below the level of detection. Eighteen (18/28) specimens were taken from six tumours (samples 563, 376, 604, 581, 607, and 1,285). **(B)** 10 GBM stem-like cell lines (designated as GCGR-E) derived from resected tissues, two fetal brain cell lines (designated as GCGR-NS), an established GBM cancer cell line (U251), and human GBM tissue # 376-1 (Tis 376-1) as positive control was immunoblotted to detect BTK protein. In all cell lines (GBMs and neural), BTK protein expression was below the level of detection. Black lines indicate a lane was removed.

expressed in GBM cancer cells, and as such, BTK inhibitors are being developed in clinical trials for GBM patients (NCT03535350). Therefore, we set out to define the cell type(s) in which BTK protein is expressed in GBM tumour tissue to better inform an interpretation of this emerging clinical trials data and better stratify patients.

Cancer tissue lysate contains many cell types, including tumour cells, normal tissue architectural cells, and immune infiltrates. Our

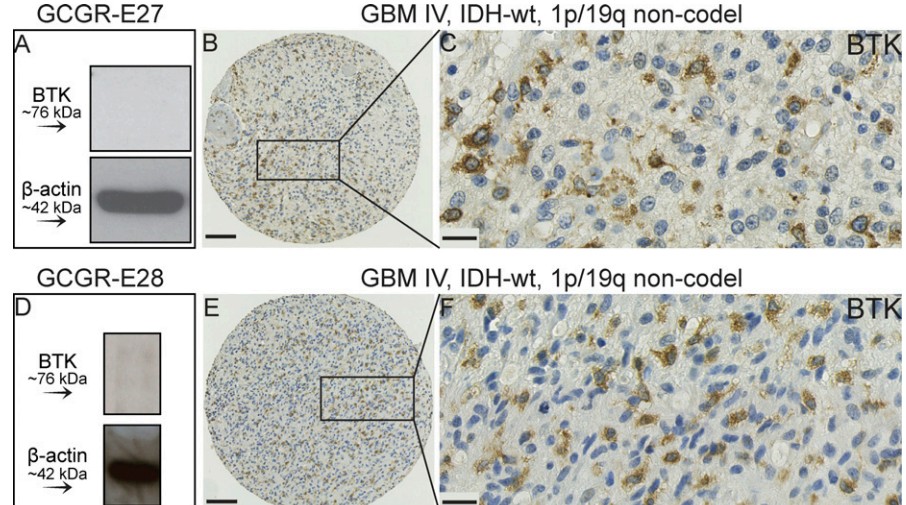

**Figure 6. Representative GBM cores showing the detection of Bruton tyrosine kinase (BTK) protein expression in two GBM samples.**

**(A, B, C)** Tumour tissue used to generate GBM stem-like cell line GCGR-E27. **(A)** BTK protein was below the level of detection in the derived GBM stem-like cell line GCGR-E27. **(B, C)** BTK protein was expressed in the tumour tissue (IHC). **(D, E, F)** Tumour tissue used to generate GBM stem-like cell line GCGR-E28. **(D)** BTK protein was below the level of detection in the derived GBM stem-like cell line GCGR-E28. **(E, F)** BTK protein was expressed in the tumour tissue (IHC) compared to the patient-derived GBM stem-like cell line. Whole immunoblots are shown in Fig S4. Scale bars (B, E) = 100 μm; (C, F) = 20 μm in. wt, wild-type; non-codel, non-codeleted.

observation that BTK expression was lower in GBM cell lines compared to GBM tissue samples may reflect that its expression is in a non-cancer cell type or the GBM stem-like cells from a BTK-negative tumour niche. In addition, BTK expression is classically associated with cell lineages of the haematopoietic system, such as B-lymphocytes, monocytes, and macrophages (46, 47). However, previous studies reported very weak infiltration of B-lymphocytes within GBM, suggesting that B cells are an unlikely site for the observed BTK activity (48). Other immune infiltrates, such as macrophages or microglia, within the GBM tissue may instead contribute to the observed BTK activity in GBM tumour tissue rather than GBM-derived cancer cells.

We first used immunohistochemistry from formalin-fixed paraffin-embedded (FFPE) material to assess BTK protein expression in a glioma tissue microarray (TMA) alongside immune cell markers, such as CD163. SOX2 was used as a marker of GBM cancer stem-like cells. The TMA consisted of 127 WHO grade II–IV glioma samples, including GBM grade IV, anaplastic astrocytoma (AA) grade III, anaplastic oligodendroglioma (AO) grade III, diffuse astrocytoma (DA) grade II, and oligodendroglioma (O) grade II. The clinical and molecular information for each of the 127 glioma tissue samples used to generate the TMA are summarised in Table 1.

We evaluated BTK protein expression by quantifying the percentage of BTK-positive staining in each core using QuPath, an open-source quantitative pathology and bioimaging software (49). We included all grades (Table 1) because lower grade tumours are stratified locally with respect to isocitrat dehydrogenase (IDH) mutation and we thought it would be important to stain for BTK in relation to all grades of GBM, to understand BTK expression states in relation to our library of GBM tissue including clinical indicators of the patients. The highest median of BTK expression was in WHO grade III and grade IV relapsed cases, followed by WHO grade II cores, and finally WHO grade IV primary cases, at 38%, 38%, 32%, and 30%, respectively (Fig 7). No statistical significance was observed

between the different grades. However, it is interesting to note that BTK protein is largely nuclear in grade II (Fig S5) and grade III (Fig S5) samples, whereas it is largely cytosolic in grade IV (Fig 6B, C, E, and F). Thus, the biological regulation of BTK appears quite distinct between the tumour grades which might have implications for the use of BTK kinase inhibitors as potential therapeutics.

We compared BTK protein expression in serial FFPE-derived sections against expression of immune cell markers in various glioma tissue samples using antibodies to macrophage cells (CD163) and against the glioma cancer stem-like cell marker (SOX2) (representative images of various gliomas are shown in Figs 8–11 and S5–S9). We observed several patterns of staining. For example, within the same serial GBM section, we observed regions in which most cells which were positively stained with BTK, negatively with SOX2 and negatively for CD163. This is an example of a field of cells which we consider to be BTK+/SOX2–/CD163– (Figs 8A, D, and G and S6). Elsewhere, there were tumour regions where SOX2-positive tumour cells appeared to be mostly BTK-negative. This is an example of a field of cells which we consider to be BTK–/SOX2+/CD163– (Fig 8B, E, and H). Given that our SOX2-positive GSC lines are BTK-negative, we speculate that the cell models we used ex vivo are representative of these dominant SOX2-positive cells in-vivo, with respect to BTK expression. Thus, these two examples demonstrate a striking heterogeneous and inverse expression between SOX2 and BTK proteins in the same tumour.

We next examined BTK expression in relation to CD163 expressing macrophages. CD163-positive cell fields in the same serial GBM section were typically BTK-negative and SOX2-negative (Fig 8C, F, and I). In addition, we observed an example of a primary IDH-mutant GBM case that exhibited a high BTK and SOX2 expressions (Fig 9A–D, respectively), but a low CD163 labelling (Fig 9E and F). However, we observed high SOX2 expression in another GBM case (Fig S7), whereas BTK and CD163 expressions were very low (Fig S7). Together, our GBM data suggest that BTK protein may be expressed

**Table 1.   Summary of histopathology, age, gender, and IDH mutation status information for 127 glioma TMA.**

| | Primary | | | | | | | Recurrence | | | |
|---|---|---|---|---|---|---|---|---|---|---|---|
| | Grade II | | | Grade III | | Grade IV | | Grade III | | Grade IV | |
| | DA | O | GA | AA | AO | GBM | Other | AA | AO | GBM | Other |
| Cores (n) | 3 | 10 | 1 | 7 | 4 | 66 | 4 | 1 | 1 | 29 | 1 |
| Age (yr) | | | | | | | | | | | |
| <50 | 3 | 4 | 1 | 5 | 4 | 14 | 0 | 1 | 1 | 10 | 0 |
| ≥50 | 0 | 6 | 0 | 2 | 0 | 52 | 4 | 0 | 0 | 19 | 1 |
| Gender | | | | | | | | | | | |
| Male | 1 | 4 | 1 | 5 | 2 | 36 | 3 | 1 | 0 | 20 | 0 |
| Female | 2 | 6 | 0 | 2 | 2 | 30 | 1 | 0 | 1 | 9 | 1 |
| IDH | | | | | | | | | | | |
| WT | 1 | 2 | 0 | 2 | 1 | 56 | 3 | 1 | 0 | 25 | 1 |
| MUT | 2 | 8 | 1 | 4 | 3 | 7 | | 0 | 1 | 4 | 0 |
| NOS | 0 | 0 | 0 | 1 | 0 | 3 | 1 | 0 | 0 | 0 | 0 |

DA, diffuse astrocytoma; O, oligodendroglioma; GA, gemistocytic astrocytoma; AA, anaplastic astrocytoma; AO, anaplastic oligodendroglioma; GBM, glioblastoma; Other, gliosarcoma and giant cell GBM.

**Figure 7. Box plots of Bruton tyrosine kinase protein expression in a glioma TMA from 127 patients stratified by tumour grade.**
There was no significant difference in Bruton tyrosine kinase expression between the examined grades of glioma (Mann–Whitney test). World Health Organization grade II tumours included DAs and Os; whereas World Health Organization grade III gliomas included AAs and AOs. Boxes represent 25th–75th percentiles, the middle bar identifies the median. Pri, primary; rec, recurrent.

in SOX2-positive tumour cells, CD163-positive macrophages, and a third population of cells which needs further characterisation.

Although immunohistochemistry in serial FFPE slices can allow us to gauge the general correlation, or lack thereof, in the co-expression of BTK with either SOX2 or CD163, this method does not define whether individual cells can co-express the targets. Co-immunofluorescence (co-if) was therefore performed to further examine the extent of BTK, SOX2, and CD163 co-expression within individual cells in GBM and a low-grade glioma (Figs 10, 11, S8, and S9). In GBM, approximately 25% of DAPI-positive cells did not express BTK or SOX2, 15% expressed BTK but not SOX2, 48% expressed

SOX2 but not BTK, and 12% expressed both BTK and SOX2 (Fig 10A–F). Our data suggest that higher grade gliomas do not co-express BTK and SOX2 extensively and we speculate that the GSC models we use ex vivo, which are typically SOX2+/BTK– as shown in Fig 6 are most likely representative of the dominant cancer cell type in the tumour tissue which are SOX2+/BTK– (48%, Fig 10F). For comparison, in the lower grade glioma, the percentage of BTK and SOX2 co-expression was much higher (Figs 11A–F and S8); however, patient-derived cell models from this lower grade were not available to compare with tissue SOX2 and BTK co-expression.

We next assessed the extent of co-expression of BTK and CD163 antibodies in a high-grade glioma (Fig S9). We found that BTK was co-expressed with CD163 cells (Fig S9 pink arrows), nevertheless, we observed BTK-positive cells that were CD163-negative and vice versa (Fig S9, yellow arrows and Fig S9, white arrows). We also evaluated BTK co-expression with another microglia marker (50), CD68, using co-IF on two different GBM sections (Fig S10). As expected, we observed co-staining between BTK and CD68 in many cells, but we also found in the same field other cells that were BTK+/CD68– and vice versa (Fig S10). However, using another GBM section, there was less co-expression between BTK and CD68 (Fig S10). These data are consistent with observations of BTK expression as a function of CD163 status; BTK can be expressed from CD68-positive and/or CD68 negative cells. However, our data highlight the pronounced heterogeneity of BTK protein expression in distinct cell types including cancer stem-like cells (SOX2), TAMs (CD163) and microglia (CD68). These data reinforce the need to study these cell populations in future to better inform the stratification of patients that might undergo BTK kinase inhibitor therapy.

### BTK expression in GBM associated with overall survival

Because CD163 expression predicts poor prognosis in GBM (51), we next used the immunohistochemical expression data of BTK (Figs 7, 9, and S5–S7) to determine whether BTK expression correlates with prognostic indicators, and whether it is similar to or distinct from CD163 as a predictor of poor prognosis. We examined the

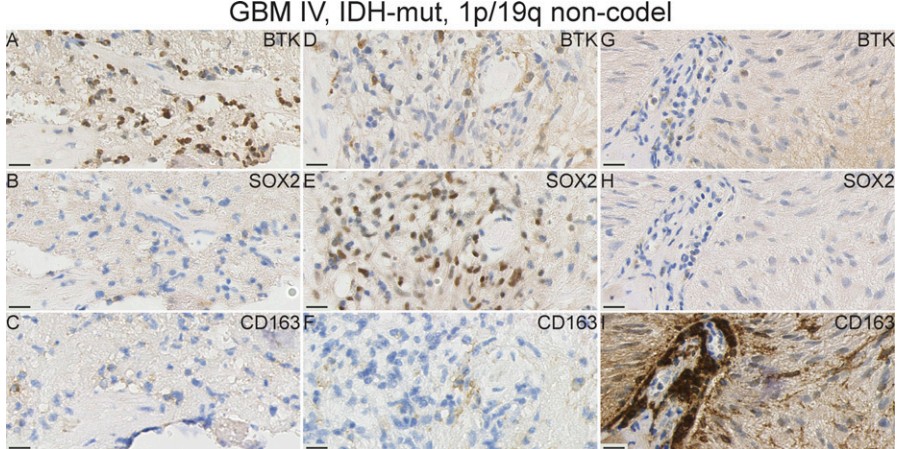

**Figure 8. IHC staining with Bruton tyrosine kinase (BTK), SOX2, and CD163 antibodies in a primary GBM case illustrates the pronounced heterogeneity of expression within one tissue sample.**
**(A, D, G)** Area with BTK+/SOX2–/CD163– expression. (Magnified region shown in Fig S6). **(B, E, H)** Area with BTK–/SOX2+/CD163– expression. **(C, F, I)** Area with BTK-/SOX2–/CD163+ expression. **(A, B, C, D, E, F, G, H, I)** tumour tissue stained with BTK, (D, E, and F) tumour tissue stained with a tumour marker SOX2, (G, H, I) tumour tissue stained with a macrophage marker CD163. Scale bars = 20 $\mu$m. Rec, recurrent; wt, wild-type; non-codel, non-codeleted.

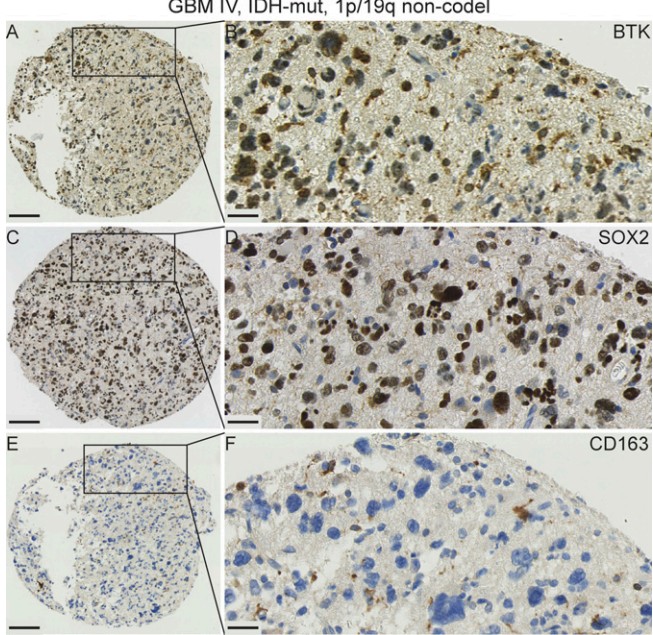

GBM IV, IDH-mut, 1p/19q non-codel

**Figure 9.  A panel showing representative IHC images from a primary GBM IV, IDH-mut, and 1p/19q non-codel.**
**(A, B)** Bruton tyrosine kinase protein expression is present in many cells, some of which are tumour cells. **(C, D)** SOX2 protein expression is detected in the majority of tumour cells nuclei. **(E, F)** CD163 expression highlighting a few scattered macrophages but the cancer cells are negative for CD163. Scale bars (A, C, E) = 100 $\mu m$; (B, D, F) = 20 $\mu m$. Mut, mutated; non-codel, non-codeleted.

relationship between BTK protein expression and overall survival time for 69 patients with primary GBM (grade IV) represented in the TMA using Kaplan–Meier survival analysis. Based on assessment of the total area of BTK expression in each tissue core, higher

expression of BTK protein significantly correlated with longer survival times (Fig 12).

Furthermore, we assessed BTK protein expression in the TMA after adjusting for other clinical data including age, KPS, and TMZ treatment. Cox's proportional hazards analysis showed that BTK expression was associated with more favourable overall survival (HR: 0.985, 95% CI: 0.972–0.988, $P$ = 0.026), chemotherapy (TMZ) treatment (HR: 0.383, 95% CI: 0.177–0.827, $P$ = 0.015) and KPS > 80 (HR: 0.509, 95% CI: 0.279–0.927, $P$ = 0.027) (Table 2). Whereas IDH-wt was the only significant negative prognostic factor for overall survival (HR: 2.700, 95% CI: 1.056–6.900, $P$ = 0.038) (Table 2). Together the KM survival curve and Cox regression analysis highlights the importance of BTK in GBM as a prognostic indicator and could have particular implications for understanding the clinical role of BTK within GBM and the utility of BTK inhibitors.

## Discussion

This study identifies a novel relevance of BTK towards GBM molecular biology. We have shown that: (1) BTK protein expression predicts good prognosis in GBM; (2) BTK is not expressed in state-of-the art GBM stem-like cell lines, but can be expressed in mixed cancer and immune cell populations within matched GBM tissue; and (3) because BTK inhibitors is already being planned for clinical trials, developing better ex vivo cell cultures that recapitulate BTK protein heterogeneity in tissue (Fig 8) will better inform the use of BTK inhibitors either as a type of immunotherapy and/or cancer cell therapy.

Kinase inhibitors represent one of the most dominant drug types currently available for use in human cancers (52). To extend our understanding of kinomes within the in-vivo tumour environment, we compared the bioactive kinome in lysates of GBM tissue and

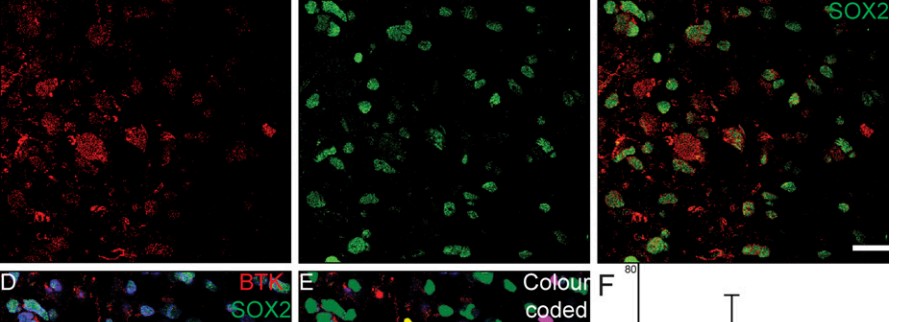

GBM VI, IDH-mut, 1p/19q non-codel

**Figure 10.  Pipeline for quantification of immunofluorescently co-labelled Bruton tyrosine kinase (BTK)/SOX2 cells in a high-grade glioma (example of GBM VI, IDH-mut, and 1p/19q non-codel) showing.**
**(A)** BTK staining (in red). **(B)** SOX2 staining (in green). **(C)** BTK and SOX2 co-staining. **(D)** BTK and SOX2 co-staining with DAPI. **(E)** Color-coded mask for each cell marker; BTK: red, SOX2: green, BTK/SOX2: yellow and DAPI: pink. This image was used for manual counting of co-expressed cells. **(F)** Boxplot graphs showing the manual counting of 12 images for each marker; percentages indicate median of the 12 images. Scale bars = 20 $\mu m$. Mut, mutated; non-codel, non-codeleted.

**Figure 11. Pipeline for quantification of immunofluorescently co-labelled Bruton tyrosine kinase (BTK)/SOX2 cells in a low -grade glioma (example of O II, IDH-mut, and 1p/19q codel) showing.** **(A)** BTK staining (in red). **(B)** SOX2 staining (in green). **(C)** BTK and SOX2 co-staining. **(D)** BTK and SOX2 co-staining with DAPI. **(E)** Color-coded mask for each cell marker; BTK: red, SOX2: green, BTK/SOX2: yellow and DAPI: pink. This image was used for manual counting of co-expressed cells. **(F)** Boxplot graphs showing medians for each marker of 12 manually counted images. Scale bars = 50 µm. O, oligodendroglioma; mut, mutated; codel, codeleted.

patient-matched cell lines using a cell-free kinomics platform. The main aim was to ask whether the key cancer cell models we use ex vivo reflect the GBM tissue kinome and/or whether additional ex vivo models need to be developed to better represent the GBM tissue kinome. The GBM tumour tissue we examined exhibited higher cell-free kinase activity compared to patient-matched cell lines. We identified a dominant kinase activity associated with BTK within GBM tissues but not in GBM cell lines. Kinome analysis on a panel of 31 GBM tissue lysates confirmed high BTK activity.

Immunoblotting showed that BTK protein expression was detectable in GBM tissue, whil in GBM stem-like cell lines BTK protein was below the level of detection. Further, BTK was found to be co-expressed to differing degrees with SOX2, CD163, or CD68, indicative of both tumour and immune cell expression of the kinase. However, most BTK-positive protein expression in GBM tissue was not linked to either SOX2-positive cells, CD163-positive cells or CD68-positive cells. As our GSC models grown in-vitro are typically SOX2+/BTK−, this indicates that these in-vitro cell models are representative of the dominant SOX2+ cancer cell population in-vivo (Fig 10). However, to better reconstitute the heterogeneity of GBM tissue and the tumour microenvironment we would propose to develop cancer cell lines that are BTK+/SOX2+ and BTK−/SOX2+, along with BTK+/CD163+ and BTK−/CD163+ immune cell populations. Such complex cell culture models might better define how BTK inhibitors impact on cancer regression or BTK inhibitor resistance.

Since the 1950s when the HeLa cell line was introduced (53), tumour-derived cell lines have formed the backbone of in-vitro cancer research and drug discovery, providing detailed insights into tumour molecular biology and pathological signalling pathways. However, although our knowledge about cancer has improved, the true utility of cancer cell lines as experimental models for developing clinically useful therapies is less clear (54, 55, 56, 57, 58, 59, 60, 61). In-vitro drug screening has generally failed to identify novel compounds that have subsequently successfully translated into the clinic to treat glioma. Tumour tissue is more complex than individual tumour cells in culture. Glioma cell lines cultured in optimum growing conditions, normoxic oxygen concentrations, with full nutritional support, and with no immune cell interactions, are unlikely to fully reproduce the complex tumour environment within the brain and may not predict drug responses where key immune cell populations can synergize in the tumour milieu in-vivo.

**Figure 12. Kaplan-Meier survival curve for Bruton tyrosine kinase (BTK) protein expression in primary GBM cases within the TMA.** Patients were split into two groups based on median BTK expression. Cases with BTK expression ≥30% were assigned to a high group, and cases with BTK expression <30% were assigned to a low group.

**Table 2. Multivariate Cox regression model for Bruton tyrosine kinase (BTK), combined with demographic and clinical factors affecting overall survival in the TMA GBM primary patients.**

| Variable | P.value | HR (95% CI) |
|---|---|---|
| Gender | 0.379 | 0.778 (0.445–1.360) |
| Age | 0.764 | 1.004 (0.977–1.033) |
| KPS | **0.027** | 0.509 (0.279–0.927) |
| Subtotal resection | 0.337 | 1.444 (0.682–3.059) |
| Biopsy | 0.136 | 2.180 (0.783–6.069) |
| Radiotherapy (RT) | 0.528 | 1.501 (0.425–5.295) |
| Chemotherapy (TMZ) | **0.015** | 0.383 (0.177–0.827) |
| IDH status | **0.038** | 2.700 (1.056–6.900) |
| BTK expression | **0.026** | 0.985 (0.972–0.998) |

All variables are categorical except age and BTK expression were continuous. Gender: male to female; RT and TMZ: untreated to treated; IDH status: wild type to mutant; Karnofsky performance scale (KPS): >80 to ≤ 80. Extent of resection: gross total resection is the reference.

Studies that have compared pathway-specific gene expression of tumour cell lines, tumour tissue samples and normal tissues samples reported that most of the cell lines exhibited up-regulation of genes related to cell cycle progression and metabolic pathways (62, 63). Genes associated with cell adhesion molecules and membrane signalling proteins were down-regulated compared with tumour and normal tissue samples. Further evidence that tumour-derived cell lines do not recapitulate the primary tumour tissue comes from a large-scale genomic study that found most solid tumours and blood cancers cell lines clustered together rather than with their primary tumour site (54, 64). Similarly, GBM cell lines cluster together, separately from the GBM tissue samples, with up-regulation of survival-related genes in the cell lines (65). The differences in kinome activity between GBM tissue and cell lines that we observed are consistent with these genomic studies.

The differences in kinome activity have several potential explanations. First, in-vitro culture conditions may select for cells that are not representative of the tumour tissue cancer cell bulk. Second, the tumour tissue lysate is a homogenate of immune, endothelial, and stromal cells, in addition to tumour cells, and the observed kinase activity may relate to those non-tumour cells. The kinase activity of these non-tumour cells likely impacts on the druggable tumour cell proteome itself. Our observations using IHC suggest that some of the observed differences between the in-vitro and tissue analysis comes from the glioma microenvironment. This is not surprising because the microenvironment is not recapitulated in culture. For example, in-vivo oxygen concentration is reported to be less than 2% (66), not the 21% oxygen concentration used in culture. Furthermore, unlike in tissue, cells in culture are not generally grown in three dimensions, which might impact on cell-density effects including kinase signalling pathways.

The dominating kinase target that we identified in tissue samples was BTK. BTK is a non-RTK belonging to the TEC kinase family (46). Mutations affecting the BTK gene lead to a disease called X-linked agammaglobulinemia (XLA) (67). It is a vital kinase for B-cell proliferation,

survival and differentiation (46). BTK has been associated with blood malignancies (68), and recently with several solid tumours including glioma (reviewed in reference 35). Clinical trials using BTK inhibitors are known to target different cancer including GBM (NCT03535350), prostate (NCT02643667), oesophagogastric (NCT02884453), refractory colorectal (NCT03332498), and immune cell populations in COVID-19 disease (NCT04382586, NCT04439006, and NCT04346199). We found that BTK is the proposed kinase for five of the most abundant peptide substrates in GBM tumour tissues relative to tumour cell lines. A recent study used samples from two GBM patients for kinomics analysis and reported that BTK kinetic activity and BTK-related transcription factors were higher within the GBM tumour (69). These studies suggest that BTK is a dominant kinase in cancer cells within GBM and that inhibiting BTK might have a therapeutic impact. By contrast, our study demonstrated that although BTK is a dominant bioactive kinase within GBM tissue, neither patient-derived cancer stem-like cell lines nor SOX2-positive cancer cells within GBM tissue express detectable or high levels of BTK protein, that is, 12% of cells co-express SOX2 and BTK (Fig 10). Therefore, a question emerges on the value of targeting BTK within tumour tissue and/or how to manage patient stratification to maximize the exploitation of BTK as a therapeutic target. For example, if BTK is largely expressed in non-cancer cell (e.g., immune cell) populations within GBM tissue, its inhibition with BTK targeted drugs might be considered a type of immunotherapy rather than cancer cell therapy.

In glioma, BTK is considered a cancer cell biomarker (36), although our data using co-immunofluorescence show that most of the SOX2-positive cancer cells do not express BTK (Fig 10). Using co-immunofluorescence, we found that only a small proportion of high-grade gliomas express BTK, although this was increased in lower grade gliomas. We would therefore not consider BTK as a robust glioma tumour biomarker, particularly not for high-grade IDH wild-type GBMs. Indeed, BTK protein expression predicts good prognosis, which is perhaps reflective in its higher expression in lower grades. Whether IDH-mutant gliomas express more BTK needs to be further investigated with more cases. BTK has been classically detected in the early stages of B cell development (46), and its expression has been reported from other immune cells such as macrophages (34, 46). We also observed co-expression of BTK with CD163 and CD68 in a proportion of cells from high-grade glioma tissues. However, most of the BTK-positive cells are CD163 negative. We suspect that BTK protein expression most likely emerges from cell types other than SOX2-positive glioma cancer cells, CD163-positive macrophages or CD68-positive microglia. A recent single-cell RNAseq analysis of cell types in GBM identified BTK expression largely in myeloid populations (70), thus we expect that BTK might be expressed in CD163-negative immune cell populations. Our future investigations will aim to further characterise and understand the BTK+\SOX2−\CD163− cell populations in GBM.

Finally, across glioma grades, it was interesting to note that BTK protein expression was observed in different cellular compartments; the nucleus in lower grade and/or the cytoplasm in higher grades, and this is consistent with published studies (46, 71). BTK protein is known as a cytoplasmic protein (46), but studies have demonstrated that BTK has the ability to shuttle between the nucleus and cytoplasm (46, 71). However, the role of BTK within the

nucleus in glioma needs further investigation. We demonstrated that BTK protein expression is associated with longer survival times in GBM. As BTK has been shown to serve as tumour suppressor gene (72, 73, 74, 75), this may in part explain these findings that BTK is linked to a more favourable prognosis. Recent studies have proposed a potential mechanism of the "protective" role of BTK as a tumour suppressor (76, 77, 78). BTK was shown to phosphorylate, regulate and increase p53 protein levels in the presence or absence of DNA damage as well as through phosphorylation of MDM2, inhibiting its function, and therefore disrupting MDM2 negative feedback on p53 (76, 77). Rada and colleagues further demonstrated that BTK can induce cell death and apoptosis in response to DNA damage independently of P53 by up-regulating p73 in the absence of p53 (78). These mechanisms may explain the association we observed between favourable survival and increased BTK expression. However, in contrast to our data, other reports have suggested that BTK expression predicts poor prognosis. Inhibiting or silencing BTK led to reduced cellular proliferation and migration, induced apoptosis in glioma cell lines and decreased sphere formation and tumourigenesis in xenograft mouse models (37, 44, 45).

In conclusion, our observations that GBM tissue samples have a higher, if not different, kinome compared with GBM-derived cell lines suggests that we can improve the ex vivo complexity of GBM cell models to maximize physiological relevance with respect to kinase signalling and kinase inhibitor drug screening. More sophisticated models are needed that better recapitulate the biochemical complexity of the human tumour tissue. Better understanding of tumoral heterogeneity in different cell types exhibiting kinase expression has important implications for future drug screening and development of targeted therapies.

# Materials and Methods

## Cell culture and conditions

GBM patient samples were acquired under informed consent according to local ethical approvals (LREC 115/ES/0094). Human neural samples were acquired under informed consent according to local ethical approvals (08/S1101/1). GBM stem-like cell lines (GCGR-E20, GCGR-E21, GCGR-E22, GCGR-E25, GCGR-E27, GCGR-E28, GCGR-E34, GCGR-E56, GCGR-E64, and GCGR-E65) and human neural stem-like cell lines (GCGR-NS12ST_A and GCGR-NS17ST_A) were obtained from The Glioma Cellular Genetics Resource (GCGR) (http://gcgr.org.uk/) and cultured according to a previously published method (79). G-222 and G-221 cancer cell lines were generated as described previously (80), and were cultured in Advanced DMEM/F12 medium and supplemented with B27, N2, penicillin/streptomycin mixture, L-glutamine, amphotericin B (Thermo Fisher Scientific), heparin (Sigma-Aldrich), and recombinant human EGF & FGF protein (R&D Systems). Culture passages for both cell lines from the initial tumour were 9 and 12 for G-222 and G-221, respectively. The GBM cancer cell line (U251) was obtained from Dr. Noor Gammoh and cultured in DMEM supplemented with 10% (vol/vol) FBS and 1% (vol/vol) 0.5 U/ml penicillin/500 ng/ml streptomycin (P/S). Cultured cancer cells were incubated at 37°C and 5% $CO_2$. Chemicals were obtained from Sigma-Aldrich unless otherwise indicated. Hypoxia conditions was mimicked by treatment with 10 $\mu$M deferoxamine mesylate salt at 70% confluency then seeded after 24 h; fresh media condition indicates full fresh media for 24 h before harvesting resembling more nutrients and reducing wastes, removal of growth factors Hr-EGF and Hr-FGF from the media 24 h before cells were seeded and the vehicle (control) condition (control kept in same media).

## Kinetic activity assay

The functional proteomics assay of (GBM) tissue lysates and cancer cell lines lysates were performed using the Pamstation12 platform (PamGene International) and the PTK PamChip (PamGene) containing 196 phosphopeptides as previously described (81, 82). In Pamstation12, sample lysate is pumped continuously through 3D porous ceramic membranes allowing the FITC-labeled anti-phosphotyrosine antibodies (PamGene) to bind to phosphorylated immobilized substrates (Fig 1) (83). GBM tissues and cancer cell lines were lysed with kinase lysis buffer (M-PER Mammalian Extraction Buffer, Halt Phosphatase Inhibitor Cocktail and Halt Protease Inhibitor Cocktail [Thermo Fisher Scientific]). Each of the four arrays in the PamChip was blocked with 2% BSA for 30 min (Sigma-Aldrich). Protein quantification determined using Bradford assay (84). In each of the four wells, a total protein of 5 $\mu$g from each sample was mixed with kinase buffer (PamGene); containing a fluorescent anti-phosphotyrosine antibody (PY20-FITC) and 400 $\mu$M ATP, then the mixture was loaded directly after blocking. The kinase activity was measured by the signal intensity of the FITC-antibody in spot array images taken every 5 min throughout the whole program (90 min) using a computer-controlled charge-coupled device. The evolved (PamGene) software used to generate real-time kinetics data, and then final images are taken post-wash at 10, 20, 50, 100, and 200 ms (example images of 10, 50, and 200 ms are given in Fig 2A). Images were processed, quantified, and analyzed using BioNavigator6 software (PamGene) to produce the kinetics curve for each peptide in each sample as described below.

## Kinetic analysis of the kinome

BioNavigator6 (Ver 6.3.67.0) (PamGene international) software was used to create kinetic activity post-wash curve slope using images captured in several exposure times (10, 20, 50, 100, and 200 ms), multiplied by 100 and $\log_2$ transformed to display signal in all peptides (85). Moreover, this produced a value called end-level value for each peptide. The fluorescence activity increases with cycles (time), reaching its maximum value at end-level value. The phosphorylation activity data values were fitted to a slope model using Vini-Fit app in BioNavigator6 (Pamgene) to identify the maximum change of fluorescence activity between cycles for each peptide, and then by using this slope we defined empirically that the highest rate in the change of fluorescence activity occurs at cycle 32; we called this value (Vini32). Vini-fit values have a linear relationship with end-level values (according to manufacturer instructions) as shown in Fig 2E. We used the Vini32 values for further analysis, and they were processed through a global kinase normalization method (VSN) as described previously (86). VSN is a nonlinear normalization method that fits a model in which the

variance is stabilized across the signal strength, allowing comparisons across all samples. Unsupervised hierarchical clustering of peptide signals for each sample was performed in BioNavigator and is displayed as heat maps. The hierarchical clustering was performed on both the columns and rows using Euclidean distance metrics. An internal positive control, the non-phosphorylated, non-physiological peptide named ART_004_EAIYAAP FAKKKXC was removed before the analysis. A complete list of peptides and their sequences can be found at https://www.pamgene.com/ (PamGene 2017).

## Immunoblotting

GBM surgical tissues and cancer cell lines were lysed in kinase lysis buffer or 8 M Urea buffer, 0.1 M Tris (pH 8.5). After protein concentration determined using the Bradford or BCA method, a total of 30–50 μg of total protein was loaded into SDS page acrylamide gels then transferred to nitrocellulose membrane. The membranes were blocked using 5% (wt/vol) non-fat milk powder in PBS + 0.1% (vol/vol) Tween 20 (PBST) for 1 h at room temperature then incubated with the mouse anti-BTK antibody (MBS9215577, 1:2,500; MyBio-Source), gently shaking overnight at 4°C followed by the mouse secondary mouse antibody (DAKO) for 1 h at room temperature. Loading antibody β-actin (Sigma-Aldrich) was incubated for 1 h at room temperature followed by the secondary antibody (DAKO). The immunoblots were washed three times each for 5 min with PBST between each step. Antibody binding was detected by enhanced chemiluminescence.

## Immunohistochemistry

Immunohistochemistry (IHC) staining was performed in the Pathology Department at the University of Edinburgh. Samples were sectioned from FFPE tissue blocks at a thickness of 5 μm and stained on positively charged slides (Thermo Fisher Scientific) to maximize tissue adherence. IHC was performed using a BOND III autostainer with Bond Polymer Refine Detection Kit (DS9800; Leica Biosystems) according to manufacturer instructions. Slides were blocked with Peroxide 3–4% (vol/vol) hydrogen peroxide for 5 min. Antigen retrieval was performed using Bond Epitope Retrieval Solution 1 (Citrate; pH 6.0) (AR9961; Leica Biosystems) for 20 min at 100°C. Slides were incubated at room temperature with a BTK antibody (MBS9215577, 1:100; MyBioSource), SOX2 (ab97959, 1:200; Abcam), or CD163 (NCL-L-CD163, 1:100; Leica) for 20 min. Visualisation was achieved using 3,3'-diaminobenzidine (DAB) staining and haematoxylin counterstaining (Leica Biosystems). Finally, slides were dehydrated and cleared in xylene before coverslips were applied. Three FFPE tissue blocks (GBM, non-tumour brain, and tonsil, which additionally served as a positive control) were used to optimize the BTK antibody dilutions before applying on the TMA.

## Co-immunofluorescence (Co-IF)

FFPE GBM and Oligodendroglioma blocks were cut into 5 μm sections in the Pathology Department at the University of Edinburgh. Immunofluorescence staining was performed as described

previously (87). Briefly, slides were dewaxed in xylene (2 × 3 min) and rehydrated through 2 × 3 min graded ethanol washes (100%, 90% and 70%) and kept in distilled water for at least 15 min. Freshly made antigen retrieval buffer (0.1 M sodium citrate buffer pH 6.0 + 0.5 ml Tween 20) was microwaved for 10 min in a pressure cooker and slides were immersed in this buffer and microwaved for further 5 min. After cooling in the antigen retrieval buffer for 20 min, sections washed once with PBS and blocked with 5% BSA + 0.2% Triton X-100 made in TBS for 1 h at RT. Slides were incubated overnight at 4°C with primary antibodies: BTK (MBS9215577, 1:100; My Bio Source), BTK (hpa002028 1:100; Sigma-Aldrich), SOX2 (ab97959 1:200; Abcam), CD163 (CD163-L-CE 1:100; Leica), and CD68 (HPA048982, 1:100; Sigma-Aldrich) diluted in TBS containing 3% BSA. After three washes with TBS + 0.2% Triton X-100, the slides were incubated for 1 h at RT with secondary antibodies (Alexa Fluor 488 A-21206; Invitrogen and Alexa fluor 594 A-21203, both at 1:1,000; Invitrogen). After three 5-min washes, slides were counterstained with 4',6-diamidino-2-phenylindole (DAPI) (1:1,000 in PBS) for 3 min and mounted with Vectashield. Each Co-IF FFPE slide was coupled with a negative control (secondary antibody only) as shown in Fig S11. Images were acquired using a Leica SP5 confocal (Leica Microsystems Milton Keynes UK) using LAS-AF software. The microscope is equipped with a 405-nm diode; argon, 561- and 648-nm laser lines, three photomultiplier tubes; and one HyD GaSP detector. Images were scanned using a 63× and 100× objective. FCS images were acquired using LAS (Leica Microsystems) and processed in Symphotime 64 software using a PicoHarp 300 TCSPC module and APD detectors (Picoquant).

## TMA construction

The TMA was constructed as previously described (88). Briefly, FFPE tumour tissue blocks were first cut into sections and stained with haematoxylin and eosin (H&E) to confirm the presence of tumour. An H&E section from each block was marked by a neuropathologist to indicate the most representative tumour area for each patient. Tissue cores of 0.6 mm diameter were punched from the donor marked tumour tissue block using a manual tissue arrayer (Beecher Instruments Tissue Arrayer) and inserted into one empty recipient block. The recipient TMA block was cut into 5 μm thick using bench mounted microtome (Thermo Fisher Scientific) and mounted onto positively charged slides (Thermo Fisher Scientific) to maximize tissue adherence.

## TMA score and analysis

TMA slides were scanned using a Hamamatsu NanoZoomer XR Digital Pathology slide scanner at ×40 magnification. Digital images of the slides were analyzed using QuPath, an open-source pathology and bioimaging software (version 0.2.0-m7) (49). We altered a previously published method for identification and quantification of immune cells using QuPath (90). First, we identified RGB color stain vectors using the auto-detect feature of the visual stain editor within QuPath. Then we used the TMA dearrayer function to apply a grid on the whole TMA slide giving each core a unique number. All cores were examined manually at this point to exclude cores that did not contain sufficient tissue or that had

artefacts, for example, tissue folding. The watershed cell detection command was then used to detect every cell in each core separately by using a built-in cell segmentation algorithm. We next created a detection classifier via object classification function. The detection classifier was built using a Random Trees algorithm with the usage of all the 41 detection features. To train the detection classifier and make it operational, we manually draw objects (annotations) over several groups of cells of the same cell type using the Brush tool in randomly selected cores. These annotations were assigned to one of the following classes: "tumour" class which identify all stained and non-stained cells including any DAP-positive expression between cells, "other" class identify cells with nonspecific staining, for example, high background and "ignore" class which identify any incorrect detections. To enable the detection classifier to distinguish between positively and negatively stained cells, we used the Intensity feature in the detection classifier to determine the cut-off point. Each class mentioned above would have two categories: positive and negative depending on the mean DAP staining. All cells that were classed and identified as "other" were combined with "tumour" negative class using a script code applied through Script editor function and all detections in "ignore" class were excluded from further analysis.

To validate and confirm QuPath accuracy, we performed a comparison between QuPath results and manual counting. Manual counting is the gold standard for assessing protein expression in IHC. For this comparison, we calculated the relative error (RE) of the QuPath method as (QuPath-manual)/manual. The threshold for a "real difference" between QuPath and manual count was set as RE > 0.200 (20%), based on a threshold previously described to assess interobserver variability (89, 90). We evaluated 13 cores from the TMA slides for BTK protein expression representing 10% of all cores included in the TMA for this study. Cores were chosen randomly (random number generator) and proportionally to tumour grade; two cores were grade II, two cores were grade III, and nine cores were GBM. All negatively stained and positively stained cells were counted along with any positive staining out-with cells. The RE was calculated for the positive count, the negative count, the total count, and the percentage positive expression. Summary of both methods appear in Table 3. The RE was > 20% in three cores (Table 3, red values). Further analysis demonstrated that the real difference (RE > 20%) affected either the negative count or positive count, but not the total count, or the percentage of positive staining. Hence the effect of these three cores on the overall analysis was considered non-significant. Overall, the QuPath method proved to be highly accurate. Pearson correlation analysis of all the counted values between the QuPath method and manual counting for the 13 cores revealed very strong and highly significant association between the two methods (Table 4).

### Statistical analysis

BTK immunohistochemistry data were analyzed by SPSS (version 25). A Kruskal–Wallis test was used to evaluate the differences between the grades. The survival curve was estimated by using the Kaplan–Meier method and differences were tested by the log-rank test. One patient was not included because of unavailability of survival data. A $P$-value ≤ 0.05 was considered to be statistically significant.

## Data Availability

This study includes no data deposited in external repositories.

**Table 3.** QuPath and manual counting of positive Bruton tyrosine kinase protein staining, positive Bruton tyrosine kinase protein negative staining, total staining count, the percentage of positive staining, and the relative error (RE) between the two methods.

| Core number | Grade | Positive count | | | Negative count | | | Total count | | | Percentage of positive | | |
|---|---|---|---|---|---|---|---|---|---|---|---|---|---|
| | | QuPath | Manual | RE | QuPath | Manual | RE | QuPath | Manual | RE | QuPath | Manual | RE |
| 118 | 4 | 788 | 790 | −0.003 | 1,740 | 1,923 | −0.095 | 2,528 | 2,713 | −0.068 | 31.2 | 29.1 | 0.070 |
| 101 | 4 | 1,118 | 1,191 | −0.061 | 2,135 | 2,664 | −0.199 | 3,253 | 3,855 | −0.156 | 34.4 | 30.9 | 0.112 |
| 135 | 2 | 865 | 980 | −0.117 | 104 | 85 | 0.224 | 969 | 1,065 | −0.090 | 89.3 | 92.0 | −0.030 |
| 140 | 2 | 264 | 270 | −0.022 | 1,621 | 1,677 | −0.033 | 1,885 | 1,947 | −0.032 | 14.0 | 13.9 | 0.010 |
| 16 | 3 | 238 | 271 | −0.122 | 652 | 632 | 0.032 | 890 | 903 | −0.014 | 26.7 | 30.0 | −0.109 |
| 153 | 3 | 642 | 767 | −0.163 | 2,517 | 2,957 | −0.149 | 3,159 | 3,724 | −0.152 | 20.3 | 20.6 | −0.013 |
| 14 | 4 | 446 | 478 | −0.067 | 2,528 | 2,703 | −0.065 | 2,974 | 3,181 | −0.065 | 15.0 | 15.0 | −0.002 |
| 47 | 4 | 434 | 370 | 0.173 | 814 | 897 | −0.093 | 1,248 | 1,267 | −0.015 | 34.8 | 29.2 | 0.191 |
| 56 | 4 | 801 | 830 | −0.035 | 1,583 | 1,695 | −0.066 | 2,384 | 2,525 | −0.056 | 33.6 | 32.9 | 0.022 |
| 69 | 4 | 943 | 923 | 0.022 | 248 | 255 | −0.027 | 1,191 | 1,178 | 0.011 | 79.2 | 78.4 | 0.011 |
| 72 | 4 | 694 | 713 | −0.027 | 1,759 | 1,676 | 0.050 | 2,453 | 2,389 | 0.027 | 28.3 | 29.8 | −0.052 |
| 81 | 4 | 549 | 508 | 0.081 | 940 | 1,037 | −0.094 | 1,489 | 1,545 | −0.036 | 36.9 | 32.9 | 0.121 |
| 89 | 4 | 1,305 | 1,339 | −0.025 | 1,633 | 2,115 | −0.228 | 2,938 | 3,454 | −0.149 | 44.4 | 38.8 | 0.146 |

Red values represent the real difference between the methods. QP, QuPath; Man, manual; RE, relative error.

**Table 4. Pearson correlation analysis between the QuPath method and the manual counting methodology.**

|  | Rho | P-value |
|---|---|---|
| Positive count | 0.975 | **>0.0001** |
| Negative count | 0.984 | **>0.0001** |
| Total count | 0.989 | **>0.0001** |
| Percentage of positive | 0.992 | **>0.0001** |

Results are shown for all the 13 cores; including the positive count, the negative count, the total count, and the percentage of positive expression. Pearson correlation coefficients and *P*-values are shown.

# Supplementary Information

# Acknowledgements

We would like to thank Helen Caldwell from the pathology department at the University of Edinburgh for IHC preparation and staining. We would like to thank the members of the Advanced Imaging Resource facility at the Institute of Genetics and Cancer (IGC). We would like to thank Gillian Morrison for supply and advice on the GCGR lines. The Glioma Cellular Genetics Resource (www.gcgr.org.uk) is funded by Cancer Research UK (C157/A21992). S Al Shboul is supported by a doctoral scholarship from Hashemite University (Jordan). This work was partially supported by European Regional Development Fund - Project ENOCH (No.CZ.02.1.01/0.0/0.0/16_019/0000868) and Ministry of Health, Czech Republic MH CZ - DRO (MMCI, 00209805), the Biotechnology and Biological Sciences Research Council (BBSRC) (BB/C511599/1 and BB/J00751X/1; United Kingdom); The Edinburgh Brain Cancer Development Fund; The International Centre for Cancer Vaccine Science project is carried out within the International Research Agendas programme of the Foundation for Polish Science co-financed by the European Union under the European Regional Development Fund.

## Author Contributions

S Al Shboul: conceptualization, resources, data curation, formal analysis, validation, investigation, visualization, methodology, project administration, and writing—original draft, review, and editing.
OE Curran: conceptualization, data curation, formal analysis, supervision, visualization, methodology, and writing—original draft.
JA Alfaro: conceptualization, data curation, formal analysis, visualization, methodology, and writing—original draft.
F Lickiss: investigation and methodology.
E Nita: conceptualization, resources, investigation, and methodology.
J Kowalski: formal analysis and methodology.
F Naji: software and formal analysis.
R Nenutil: formal analysis, funding acquisition, and methodology.
KL Ball: conceptualization, formal analysis, funding acquisition, and methodology.
R Krejcir: investigation and methodology.
B Vojtesek: conceptualization, funding acquisition, and methodology.
TR Hupp: conceptualization, resources, data curation, formal analysis, supervision, funding acquisition, investigation, visualization, methodology, project administration, and writing—original draft, review, and editing.
PM Brennan: conceptualization, resources, data curation, formal analysis, supervision, investigation, visualization, methodology, project administration, and writing—original draft, review, and editing.

## Conflict of Interest Statement

F Naji was an employee of the PamGene company. All other authors declare no conflict of interest.

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
