## [Reviewer comments · Life Science Alliance]

Life Science Alliance

Kinomics platform using GBM tissue identifies BTK as being associated with higher patient survival

Sofian Al Shboul, Olimpia E. Curran, Javier A. Alfaro, Fiona Lickiss, Erisa Nita, Jacek Kowalski, Faris Najj, Rudolf Nenutil, Kathryn L. Ball, Radovan Krejcir, Borivoj Vojtesek, Ted R. Hupp, and Paul M Brennan

DOI: <https://doi.org/10.26508/lsa.202101054>

Corresponding author(s): Sofian Al Shboul, Hashemite University and Paul Brennan, Centre for clinical brain sciences

Review Timeline:	Submission Date:	2021-02-12
	Editorial Decision:	2021-06-09
	Revision Received:	2021-09-06
	Editorial Decision:	2021-09-21
	Revision Received:	2021-09-27
	Accepted:	2021-09-27

Transaction Report:

June 9, 2021

Re: Life Science Alliance manuscript #LSA-2021-01054-T

Sofian Al Shboul
MRC Institute of Genetics & Molecular Medicine
Crewe road
Edinburgh EH4 2XU
United Kingdom

Dear Dr. Al Shboul,

Thank you for submitting your manuscript entitled "A kinomics platform in glioblastoma identifies Bruton Tyrosine Kinase as being associated with higher patient survival" to Life Science Alliance. The manuscript was assessed by expert reviewers, whose comments are appended to this letter. We invite you to submit a revised manuscript addressing the Reviewer comments.

Thank you for this interesting contribution to Life Science Alliance. We are looking forward to receiving your revised manuscript.

Sincerely,

Eric Sawey, PhD
Executive Editor
Life Science Alliance
<http://www.lsa-journal.org>

- A letter addressing the reviewers' comments point by point.
- An editable version of the final text (.DOC or .DOCX) is needed for copyediting (no PDFs).
- High-resolution figure, supplementary figure and video files uploaded as individual files: See our detailed guidelines for preparing your production-ready images, <https://www.life-science-alliance.org/authors>
- Summary blurb (enter in submission system): A short text summarizing in a single sentence the study (max. 200 characters including spaces). This text is used in conjunction with the titles of papers, hence should be informative and complementary to the title and running title. It should describe the context and significance of the findings for a general readership; it should be written in the present tense and refer to the work in the third person. Author names should not be mentioned.

B. MANUSCRIPT ORGANIZATION AND FORMATTING:

Reviewer #1 (Comments to the Authors (Required)):

In their manuscript "A kinomics platform in glioblastoma identifies Bruton Tyrosine Kinase as being associated with higher patient survival" Curran and their colleagues showed that using a 'functional proteomics' screen, they were able to identify a specific dominant kinase activity namely, Bruton Tyrosine Kinase (BTK) in GBM tumor tissue that was absent/lost in the same patient matched GBM lines derived from these tumors. The authors further demonstrated that expression of BTK is not confined to a specific population of cells but is expressed in three different groups of cell types, and it also based on the type of tumor tissue examined. They also added that in contrast to data reported from earlier studies, BTK is associated with better patient prognosis.

This is an interesting finding as it highlights the importance of further improving on ex-vivo models for GBM that better recapitulate the original human tumor from which they were derived. Loss of this complexity or gain of new alterations owing to in vitro culture conditions, impairs faithful utility of

drug screening and targeted therapeutic candidates in patients. Though, this is an important finding there are some major points that need to be addressed in this manuscript before its acceptance:

Major points:

Results:

1. The authors should clarify the passages of the primary GBM lines derived from G-221 and G-222 used in Figure 2. This is a crucial point since it is known that the time these lines are maintained in culture to a great extent influences their genetic/proteomic profiles.
2. In Fig 2A, what does control mean? The legend states it is G-222 cell line, but the figure has no mention of it. And the figure should be reorganized with the G-222 tissue represented as 1, followed by the G-222 cell line since 3-5 are compared with the G-222 tissue and it is hence easy to follow.
3. Figure 2B and 2C are redundant. Also, Fig 2B is very hard to understand. There is no key for the heatmap. And why are the comparisons of other media conditions done with standard media condition and not the tissue sample? The changes as shown in Fig 2B doesn't correspond with the images shown in Fig 2A. In the text, it is stated that the hypoxia growth condition, showed differences when compared with standard media. However, neither Fig 2A/B show this. If anything, #5-with addition of fresh media shows the most pronounced difference when compared with standard media (based on both Fig 2A&B). Fig 2B should be removed and Fig 2C which is more informative and corresponds with the message of the result should be retained. However, a key for the colors in Fig 2C needs to be added.
4. In Fig 2D, red and blue signify differently than the rest of the Fig 2. Again, very hard to interpret, until you read the legend. Use colors different from heatmaps, as this figure is not a heatmap necessarily since blue doesn't illustrate lower phosphorylation.
5. Figure 3- Please add a key for the heatmaps. Higher and Lower are very vague ways to explain the data. In Figure 3B, there are no labels for the x& y axes. Where is BTK in the volcano plots; since the authors state in results following Fig 3 that BTK is highly expressed in tumor tissue when compared with the cells.
6. The stratification by the authors that led to the selection of BTK as the dominant kinase, is not well explained. The stratification based only on G-222 also looks weak. Is BTK not a dominant kinase in G-221? This part of the manuscript leading to the author's focus on BTK needs further elaboration.
7. Figure 4C, has no axes labels. To understand this figure, the authors should highlight some relevant peptides. In Figure 4D, why is the expression of the control peptide ART003 so varied between the samples? This makes the interpretation of the variance observed among the other peptides questionable. The expression of the peptides should be normalized to the control peptide expression within each sample. Additionally, showing expression of peptides in matched lines with at least a few candidates will be helpful. Also, querying for candidate peptides based on only G-222 is a leap and hence should be corroborated with more lines.
8. The authors used CD163 as a candidate macrophage marker in GBM tissues. The importance of CD163 as a macrophage marker in GBM is not well established (only few recent reports show its significance and CD163 was shown in earlier reports to be expressed by tumor cells as well- Chen et al. Oncogene 2019). Other markers should be explored. Additionally, microglia markers should be co-stained with BTK. The authors strongly believe that the expression of BTK is present in non-tumor cells though the efforts to identify the cell identity is not explored as efficiently. In Fig 8, it will be useful to show insets with higher magnification since the authors' claims on the localization of BTK is not clear from the images provided. Also, the authors state that there might be a population that exhibits BTK colocalization with CD163. However, from the data provided, there is no clear

correlation between the expression of CD163 and BTK (very few co-stained cells). Better quality images should be provided, or this section of the results should be reanalyzed. The identity of the BTK positive cells in the tumors is still elusive and the authors should definitely address this.

9. The authors should provide a reasoning for the switch of analyzing BTK expression from high to low grade gliomas. According to Figure 7, the expression of BTK was present in all grades of GBM and there were no statistical differences in the expression between the grades. The initial kinome profiling was done in Grade IV GBM tumors that led to the identification of BTK as a candidate kinase. Hence, this should be elaborated on clearly.

10. Overall, the relevance of BTK still is not explained well. The relevance of the study is understood, but the relevance of BTK with respect to GBM is still questionable and is not addressed in this study. This should be addressed in the discussion.

Minor points:

1. In the second paragraph of the Introduction, the author states: "Therapy failure may be a consequence of RTK intratumoural heterogeneity or may reflect that the importance of these RTKs is different in tumour tissue compared to the tumour-derived cell lines on which the relevant preclinical work was undertaken". This is a very difficult statement to understand. The first part is logical; however, the second part is not true. Failure of therapies in vivo occurs not due to differences in the importance of the targeted RTKs but due to adaptations gained by the tumor and switch to other survival pathways.

Additional points:

1. The data in the paper is poorly organized (Especially in Figure 2). The chronology of the text doesn't align with the figures and is very hard to follow).
2. There are no keys for any heatmaps. This information is available throughout in literature for these kind of data presentations.
3. The legends should include all the information present in the figure. For example, in Figure 4, the meaning of "peptide ranks" is not included in the legend.

06 September 2021

Re: Life Science Alliance manuscript #LSA-2021-01054-T

Dear Dr Sawey

Please find attached a revised paper entitled “A kinomics platform in glioblastoma identifies Bruton Tyrosine Kinase as being associated with higher patient survival” to be considered for publication in “Life Science Alliance”. Below is the original letter, the reviewers responses, and our responses (in red).

We look forward to hearing on the suitability of our work for publication in your Journal.

Sincerely yours,

Sofian Al Shboul

University of Edinburgh
The Institute of Genetics & Cancer (IGC)
Crewe road south
Edinburgh EH4 2XU
United Kingdom

Reviewer #1 (Comments to the Authors (Required)):

In their manuscript "A kinomics platform in glioblastoma identifies Bruton Tyrosine Kinase as being associated with higher patient survival" Curran and their colleagues showed that using a 'functional proteomics' screen, they were able to identify a specific dominant kinase activity namely, Bruton Tyrosine Kinase (BTK) in GBM tumor tissue that was absent/lost in the same patient matched GBM lines derived from these tumors. The authors further demonstrated that expression of BTK is not confined to a specific population of cells but is expressed in three different groups of cell types, and it also based on the type of tumor tissue examined. They also added that in contrast to data reported from earlier studies, BTK is associated with better patient prognosis.

This is an interesting finding as it highlights the importance of further improving on ex-vivo models for GBM that better capitulate the original human tumor from which they were derived. Loss of this complexity or gain of new alterations owing to in vitro culture conditions, impairs faithful utility of drug screening and targeted therapeutic candidates in patients. Though, this is an important finding there are some major points that need to be addressed in this manuscript before its acceptance:

Major points:

Results:

1. The authors should clarify the passages of the primary GBM lines derived from G-221 and G-222 used in Figure 2. This is a crucial point since it is known that the time these lines are maintained in culture to a great extent influences their genetic/proteomic profiles.

G-221 line is passage 12 and the G-222 line is passage 9 (from the original sample, derivation) and this information is added to the methods text.

2. In Fig 2A, what does control mean? The legend states it is G-222 cell line, but the figure has no mention of it. And the figure should be reorganized with the G-222 tissue represented as 1, followed by the G-222 cell line since 3-5 are compared with the G-222 tissue and it is hence easy to follow.

Fig2A is rearranged to focus on the main issue we want to emphasize; as such the modified figure only compares G-222 GBM tissue to the G-222 matched derived cancer stem-like cell line. The other raw data that contained essentially similar data to the the cell lines (except cell line under different treatments including hypoxia, growth factors, etc) was removed but, the data was still included in Figure 2B as a heat map. We think this re-arrangement makes the data more relatable.

3. Figure 2B and 2C are redundant. Also, Fig 2B is very hard to understand. There is no key for the heatmap. And why are the comparisons of other media conditions done with standard media condition and not the tissue sample? The changes as shown in Fig 2B doesn't correspond with the images shown in Fig 2A. In the text, it is stated that the hypoxia growth condition, showed differences when compared with standard media. However, neither Fig 2A/B show this. If anything, #5-with addition of fresh media shows the most pronounced difference when compared with standard media (based on both Fig 2A&B). Fig 2B should be removed and Fig 2C which is more informative and corresponds with the message of the result should be retained. However, a key for the colors in Fig 2C needs to be added.

As in point 2, we agree that the original layout of the figures was difficult and confusing for readers. Since this figure highlights a key point (that the tissue kinome is more active than cell derived kinome), the figure was simplified to include:

1. 2A tissue vs cell line including the raw data so readers can visualize what the phospho-peptide data looks like.
2. 2B the heat map showing how changing the conditions of the cell line shows difference to the control in terms of the kinome--but the cell lines kinomes do not resemble the tissue kinome under any conditions,
3. 2C; comparing another patient's kinome using GBM tissue vs matched cell line.
4. 2D and E. an example of the kinetic analysis so readers can understand how the rate data was generated.

4. In Fig 2D, red and blue signify differently than the rest of the Fig 2. Again, very hard to interpret, until you read the legend. Use colors different from heatmaps, as this figure is not a heatmap necessarily since blue doesn't illustrate lower phosphorylation.

As above, this was all rearranged for clarity.

5. Figure 3- Please add a key for the heatmaps. Higher and Lower are very vague ways to explain the data. In Figure 3B, there are no labels for the x& y axes. Where is BTK in the volcano plots; since the authors state in results following Fig 3 that BTK is highly expressed in tumor tissue when compared with the cells.

[Figure removed by editorial staff per authors' request].

The BTK related peptides are now circled in brown, most of them are in the central region (fold change >1 in the tissue). The original volcano plot (Figure 4C) was removed.

6. The stratification by the authors that led to the selection of BTK as the dominant kinase, is not well explained. The stratification based only on G-222 also looks weak. Is BTK not a dominant kinase in G-221? This part of the manuscript leading to the author's focus on BTK needs further elaboration.

After establishing the differences between the tissue and derived cell line (Fig 2A), we aimed to identify any features that are missing from the cell models. We ranked the highest 10 peptides (according to their phosphorylation activity) from the G-222 experiment and noticed that 6 of them are associated with BTK (Fig 3 and Fig 4). These 6 peptides (Fig 4A, in pink) were higher in tissue compared to the cell line. We also identified another 12 BTK-related peptides from the peptides that were ranked 11-100 (Fig 4A and B) (This was mentioned in Fig 4 legend). In turn, BTK protein itself was elevated in tissue, when we blotted 29 GBM tissues vs 11 GBM cell lines (Figure 5).

To test the concept in another GBM derived tumour from a patient; G-221, we have only two tissue samples from this surgery so from the delta of tissue vs the control cells, we observed similar findings when we performed the same analysis for the GBM G-221 tissue and

matched cell line (Fig S1A); all BTK related peptides were higher in tissue relative to matched cell line including the same six within the top active ten peptides.

7. Figure 4C, has no axes labels. To understand this figure, the authors should highlight some relevant peptides. In Figure 4D, why is the expression of the control peptide ART003 so varied between the samples? This makes the interpretation of the variance observed among the other peptides questionable. The expression of the peptides should be normalized to the control peptide expression within each sample. Additionally, showing expression of peptides in matched lines with at least a few candidates will be helpful. Also, querying for candidate peptides based on only G-222 is a leap and hence should be corroborated with more lines.

All the BTK relevant peptides were marked with brown circles (Figure 3B, please see above). We looked at the statistically significant BTK peptides from (Fig 3B, brown circled, 18 peptides) in the G-222 pair, shown in (Fig S1B), all the 18 peptides were higher in G-221 tissue relative to G-221 cell line).

ART003 is a synthetic qualitative peptide that acts as a control for the antibody rather than to the sample. This was clarified in the figure legend as follows: “The synthetic pre-phosphorylated peptide named ART003 (EAI(phospho-Y)AAPFAKKKXC) (bottom of the row in Figure 4C) is a non-physiological control phospho-peptide incorporated by the Pamstation12® user platform that functions as a positive control as it exhibits a high basal signal due to its prior phosphorylation. As this phospho-peptide is “100%” phosphorylated, any reductions in its phosphorylation (such as indicated by the green colour in tissue 879) is likely due to phosphatase(s) from the tissue or cell lysate that de-phosphorylates the peptide. Data from this phospho-peptide was not used in any pathway assessments as it is only used as a positive control to yield a high phospho-signal in the Pamstation12®”. in addition, for the normalization, we tested several normalization methods before we used VSN. VSN is used with this kind of proteomics data [1]

In addition, as suggested above, we also include the data from another patient (G-221) as a corroborated (Fig 2C), which shows the tissue derived kinome and cell derived kinome from the same patient have a distinction similar to G-222.

8. The authors used CD163 as a candidate macrophage marker in GBM tissues. The importance of CD163 as a macrophage marker in GBM is not well established (only few recent reports show its significance and CD163 was shown in earlier reports to be expressed by tumor cells as well- Chen et al. Oncogene 2019). Other markers should be explored. Additionally, microglia markers should be co-stained with BTK.

The authors strongly believe that the expression of BTK is present in non-tumor cells though the efforts to identify the cell identity is not explored as efficiently.

In Fig 8, it will be useful to show insets with higher magnification since the authors' claims on the localization of BTK is not clear from the images provided.

Also, the authors state that there might be a population that exhibits BTK colocalization with CD163. However, from the data provided, there is no clear correlation between the expression of CD163 and BTK (very few co-stained cells). Better quality images should be provided, or this section of the results should be reanalyzed.

The identity of the BTK positive cells in the tumors is still elusive and the authors should definitely address this.

We stained BTK with CD68 in GBM tissues and provided better images of BTK/CD163, including more co-stained cells as a representative example, in (Fig S8-S9). This new data demonstrates a degree of co-expression of BTK with the microglial marker, CD68. The main point of Fig 8 (magnified region shown in Fig S5) was to show that within the same GBM tumour there were regions where BTK protein is highly expressed, where SOX2 and CD163 both were both negative. Such observation highlights, firstly, GBM cell heterogeneity and secondly, that the BTK+ cells that were not evident in cell models have relevance when examining tissue itself.

Identification of the nature of the BTK positive “immune/stromal” cells requires substantial work that is beyond the scope of this study; the original goal of our study was to determine whether cell line kinomes mimic matched tumour tissue. We agree that identification of the cell of origins containing BTK is interesting but will require live cell fractionation, of disaggregated tumour GBM tissue from surgery, and will be the focus of our next work.

9. The authors should provide a reasoning for the switch of analyzing BTK expression from high to low grade gliomas. According to Figure 7, the expression of BTK was present in all grades of GBM and there were no statistical differences in the expression between the grades. The initial kinome profiling was done in Grade IV GBM tumors that led to the identification of BTK as a candidate kinase. Hence, this should be elaborated on clearly.

This was clarified in the text; on page 16; “We included all grades (Table 1) because lower grade tumours are stratified locally with respect to IDH mutation and we thought it would be important to stain for BTK in relation to all grades of GBM to describe BTK in relation to our library of GBM tissue and clinical indicators”. The additional reason for including staining of BTK in low grade gliomas came from our observation that BTK staining in low grades was nuclear in localization compared to cytosolic localization in GBM. This was included in the text as follows: “However, it is interesting to note that BTK protein is largely nuclear in Grade II and III samples (Figure S3-S4), whilst is largely cytosolic in Grade IV (Figure 6)”. These observations will be used to drive our future work and hypothesis of nuclear BTK localization and whether this is influenced by IDH mutation status.

10. Overall, the relevance of BTK still is not explained well. The relevance of the study is understood, but the relevance of BTK with respect to GBM is still questionable and is not addressed in this study. This should be addressed in the discussion.

The manuscript has been re-written at several points to emphasise that the relevance of BTK; in particular the discussion starts off with this review.

“This study identifies a novel relevance of BTK towards GBM molecular biology. We have shown that: 1. BTK protein expression predicts good prognosis in GBM; 2. BTK is not expressed in state-of-the art GBM stem cell lines, but can be expressed in mixed cancer and immune cell populations within matched GBM tissue; and 3; because BTK inhibitors is already being planned for clinical trials, developing better *ex-vivo* cell cultures that recapitulate BTK protein heterogeneity in tissue (Figure 8) will better inform the use of BTK inhibitors either as a type of immune therapy and/or cancer cell therapy .”

Minor points:

1. In the second paragraph of the Introduction, the author states: "Therapy failure may be a consequence of RTK intratumoural heterogeneity or may reflect that the importance of these RTKs is different in tumour tissue compared to the tumour-derived cell lines on which the relevant preclinical work was undertaken". This is a very difficult statement to understand. The first part is logical; however, the second part is not true. Failure of therapies *in vivo* occurs not due to differences in the importance of the targeted RTKs but due to adaptations gained by the tumor and switch to other survival pathways.

Changed section was altered for clarity;

“Therapy failure may be a consequence of RTK intratumoural heterogeneity [2-4], or may reflect that these RTKs could have various functions/roles in tumour tissue which could be lost/altered upon deriving cell lines because of culture conditions.”

Additional points:

1. The data in the paper is poorly organized (Especially in Figure 2). The chronology of the text doesn't align with the figures and is very hard to follow).

As above, we agree that figure 2 was confusing and it has been rearranged for clarity.

2. There are no keys for any heatmaps. This information is available throughout in literature for these kind of data presentations.

Keys were included as a reference point.

3. The legends should include all the information present in the figure. For example, in Figure 4, the meaning of "peptide ranks" is not included in the legend.

“Peptides were ranked according to their phosphorylation activity; higher rank indicates higher phosphorylation rate (peptide ranks).”

This was mentioned in the legend of Fig 4

1. Hurkmans, D.P., et al., *Blood-based kinase activity profiling: a potential predictor of response to immune checkpoint inhibition in metastatic cancer*. *J Immunother Cancer*, 2020. **8**(2).
2. Aum, D.J., et al., *Molecular and cellular heterogeneity: the hallmark of glioblastoma*. *Neurosurg Focus*, 2014. **37**(6): p. E11.
3. Soeda, A., et al., *The evidence of glioblastoma heterogeneity*. *Sci Rep*, 2015. **5**: p. 7979.
4. Szerlip, N.J., et al., *Intratumoral heterogeneity of receptor tyrosine kinases EGFR and PDGFRA amplification in glioblastoma defines subpopulations with distinct growth factor response*. *Proc Natl Acad Sci U S A*, 2012. **109**(8): p. 3041-6.

September 21, 2021

RE: Life Science Alliance Manuscript #LSA-2021-01054-TR

Dr. Sofian Al Shboul
MRC Institute of Genetics and Cancer
Crewe road
Edinburgh EH4 2XU
United Kingdom

Dear Dr. Al Shboul,

Thank you for submitting your revised manuscript entitled "Kinomics platform using GBM tissue identifies BTK as being associated with higher patient survival". We would be happy to publish your paper in Life Science Alliance pending final revisions necessary to meet our formatting guidelines.

- please upload one figure per page if possible (currently one figure is split up into a few pages)
- please add the Twitter handle of your host institute/organization as well as your own or/and one of the authors in our system
- there is a name discrepancy for one of the Authors. Please correct Borek Vojtesek in manuscript file vs. Bořivoj Vojtěšek in the system
- please add your main, supplementary figure, and table legends to the main manuscript text after the references section
- Please indicate molecular weight next to each protein blot in Figure 6
- please be sure to add callouts for all panels of all figures to the main manuscript text
- please mention in the figure legend what the lines drawn on the blots indicate in Figure 5

A. FINAL FILES:

B. MANUSCRIPT ORGANIZATION AND FORMATTING:

Sincerely,

Eric Sawey, PhD

Executive Editor
Life Science Alliance
<http://www.lsjournal.org>

Reviewer #1 (Comments to the Authors (Required)):

The questions have been addressed substantially by the authors.

September 27, 2021

RE: Life Science Alliance Manuscript #LSA-2021-01054-TRR

Dr. Sofian Al Shboul
Hashemite University
Damascus Hwy
Zarqa 13133
Jordan

Dear Dr. Al Shboul,

Thank you for submitting your Research Article entitled "Kinomics platform using GBM tissue identifies BTK as being associated with higher patient survival". It is a pleasure to let you know that your manuscript is now accepted for publication in Life Science Alliance. Congratulations on this interesting work.

DISTRIBUTION OF MATERIALS:

Again, congratulations on a very nice paper. I hope you found the review process to be constructive and are pleased with how the manuscript was handled editorially. We look forward to future exciting submissions from your lab.

Sincerely,
